# Combined loss of LAP1B and LAP1C results in an early onset multisystemic nuclear envelopathy

Boris Fichtman[1], Fadia Zagairy[1], Nitzan Biran[1], Yiftah Barsheshet[1], Elena Chervinsky[2], Ziva Ben Neriah[3], Avraham Shaag[4], Michael Assa[1], Orly Elpeleg[4], Amnon Harel [1] & Ronen Spiegel[5,6]

Nuclear envelopathies comprise a heterogeneous group of diseases caused by mutations in genes encoding nuclear envelope proteins. Mutations affecting lamina-associated polypeptide 1 (LAP1) result in two discrete phenotypes of muscular dystrophy and progressive dystonia with cerebellar atrophy. We report 7 patients presenting at birth with severe progressive neurological impairment, bilateral cataract, growth retardation and early lethality. All the patients are homozygous for a nonsense mutation in the *TOR1AIP1* gene resulting in the loss of both protein isoforms LAP1B and LAP1C. Patient-derived fibroblasts exhibit changes in nuclear envelope morphology and large nuclear-spanning channels containing trapped cytoplasmic organelles. Decreased and inefficient cellular motility is also observed in these fibroblasts. Our study describes the complete absence of both major human LAP1 isoforms, underscoring their crucial role in early development and organogenesis. LAP1-associated defects may thus comprise a broad clinical spectrum depending on the availability of both isoforms in the nuclear envelope throughout life.

[1] Azrieli Faculty of Medicine, Bar-Ilan University, Safed, Israel. [2] Genetic Institute, Emek Medical Center, Afula, Israel. [3] Department of Human Genetics, Shaare Zedek Medical Center, Jerusalem, Israel. [4] Monique and Jacques Roboh Department of Genetic Research, Hadassah-Hebrew University Medical Center, Jerusalem, Israel. [5] Department of Pediatrics B', Emek Medical Center, Afula, Israel. [6] Rappaport School of Medicine, Technion, Haifa, Israel. These authors contributed equally: Boris Fichtman, Fadia Zagairy. Correspondence and requests for materials should be addressed to A.H. (email: amnon.harel@biu.ac.il) or to R.S. (email: spiegelr@zahav.net.il)

The nuclear envelope (NE) separates the cytoplasm from the nucleus in all eukaryotic cells and is structurally composed of the inner and outer nuclear membranes, nuclear pore complexes, and the nuclear lamina[1–3]. The perinuclear space is located between the inner and outer nuclear membranes and is continuous with the lumen of the endoplasmic reticulum (ER). Dozens of unique integral membrane proteins are anchored into the inner nuclear membrane and interact with lamins, the main constituents of the nuclear lamina[4,5]. Mutations in genes encoding essential protein components of the NE are known to be associated with specific human diseases collectively termed nuclear envelopathies[6,7]. Several known examples are mutations in the *EMD* gene causing Emery–Dreifuss muscular dystrophy[8], mutations in the *TOR1A* gene resulting in torsion dystonia[9], and mutations in the *LMNA* gene that results in a wide phenotypic spectrum including muscular dystrophy, cardiomyopathy, peripheral neuropathy, lipodystrophy and a unique premature aging syndrome termed Hutchinson–Gilford progeria syndrome (HGPS)[10]. Lamina-associated polypeptide 1 (LAP1) is a ubiquitously expressed protein located in the inner nuclear membrane that was first identified as three antigenically related polypeptides in rat liver NE extracts[11,12]. The rat and mouse isoforms were later designated LAP1A, LAP1B, and LAP1C and were shown to bind assembled nuclear lamins in vitro[13]. At least two functional LAP1 isoforms, namely, LAP1B and LAP1C, are known in humans and arise from a single gene designated *TOR1AIP1*[14,15]. LAP1 interacts with several proteins, including nuclear lamins, emerin, and torsinA, forming a functional network with variable expression in different tissues depending on protein content and interaction levels[16–19] (reviewed in ref. [20]). The highly similar luminal domains of LAP1 and LULL1 have both been shown to independently bind and activate torsin ATPases[21]. TorsinA interacts in the perinuclear space with components of the linker of nucleoskeleton and cytoskeleton (LINC) complex[22]. Thus, in the broader sense, LAP1 belongs to a dynamic and elaborate network of interactions spanning the perinuclear space and connecting the nuclear lamina, the NE, and the cytoskeleton.

Recessive mutations in the *TOR1AIP1* gene have been reported to result in two separate phenotypes, both arising during childhood following asymptomatic infancy, of muscular dystrophy with cardiac involvement[23,24] and a neurological phenotype dominated by dystonia and progressive cerebellar atrophy[25].

Here we report seven patients of similar ethnic background presenting at birth with a multisystemic disease dominated by profound psychomotor retardation, cataract, heart malformation, sensorineural deafness, and peculiar facial appearance associated with homozygosity for a *TOR1AIP1* loss-of-function mutation. Patient-derived fibroblasts exhibit a set of unique phenotypes that differ from the common cellular hallmarks of other nuclear envelopathies. These include reduced anti-lamin nuclear rim staining, large nuclear-spanning channels containing trapped cytoplasmic organelles, and severely impaired cellular motility.

## Results

### Clinical summary

The patients of the current study are seven individuals (six females and one male) from five separate sibships (Supplementary Fig. 1). Six of these patients originate from Arab Muslim families living in a Northern Israeli city of 50,000 inhabitants with an extremely high inbreeding rate, and another is from an Arab Muslim consanguineous family in the Jerusalem region. All patients are from Palestinian ancestry. Four patients (I-2, I-3, I-4, and II-1) already died at the ages of 8.5, 9.5, 5, and 8.5 years, respectively. The other three individuals (III-3, IV-4, and V-2) are alive and their current ages are 3.5, 3, and 6 years, respectively.

All the patients presented a distinctive phenotype with the typical features detailed in Table 1. As a rule, birth weight and head circumference were significantly low representing intrauterine growth retardation and fetal onset microcephaly. Bilateral cataract, sensorineural deafness, and significant hypotonia were already evident at birth. Heart malformations were identified at birth in four patients, including tetralogy of Fallot (I-3) and large ventricular septal defect (I-4, V-2), all requiring surgical repair. Disease course was similar in all patients, dominated by failure to gain weight as manifested by severe cachexia, muscle wasting, and dystrophic appearance (Fig. 1); evolving microcephaly; and profound global psychomotor retardation featured by the lack of attaining any developmental milestones, including social smile, the ability to roll, and to reach out for an object. Over the years, the marked infantile hypotonia (Fig. 1d) was gradually replaced by a combination of truncal hypotonia and limb hypertonia and the development of tendon contractures. Despite surgical cataract extraction within the first months of life and attempts to use hearing aids, the patients remained legally blind and deaf and therefore unable to communicate with their environment.

Late complications were seen over the years with recurrent respiratory infections, feeding difficulties, and gastro-esophageal reflux that required the insertion of gastrostomy tube in most patients (Fig. 1f). Interestingly, all the patients shared typical dysmorphic features mainly of the face including microphthalmia, deep set eyes, long philtrum, arched eyebrows, flat occiput, and high palate (Fig. 1a–c). Over the years, the patients gradually develop progeroid-like appearance. Other, less common manifestations seen in some of the patients were gastrointestinal (congenital ileal atresia, chronic constipation), renal (nephrolithiasis), and skin involvement (limb hypertrichosis, vascular reticular changes) (Fig. 1e, f).

Brain magnetic resonance imaging (MRI), when performed, demonstrated global impairment with hypoplastic and dystrophic corpus callosum, global cortical atrophy, cerebellar (mainly vermian) atrophy, and substantial delay in myelination (Fig. 1g, h) being the most common characteristics. Notably, fetal MRI was performed in one patient (I-4) at 32 weeks gestation and showed normal head circumference as well as normal cortical development and corpus callosum structure. However, cerebellar vermis and pontine size were below the third centile, suggesting intra-uterine ponto-cerebellar hypoplasia had already occurred. Collectively, the serial MR images indicate ongoing global parenchymal brain atrophy that already begins in fetal life and continues progressively thereafter resulting in severe microcephaly.

### Identification of a nonsense mutation in the *TOR1AIP1* gene.

We elected to proceed with a genetic investigation using whole-exome sequencing (WES) under the hypothesis of a recessively inherited rare allele. WES of patient III-3 and patient V-2, originating from two separate pedigrees, yielded 48.9 and 66.2 million mapped reads with a mean coverage of 85× and 100×, respectively. Following alignment and variant calling, we performed a series of filtering steps. These included the removal of variants called <7×, as well as variants that were off-target, synonymous, heterozygous, or had mean allele frequency (MAF) > 0.5% at ExAC (Exome Aggregation Consortium, Cambridge, MA; URL: http://exac.broadinstitute.org). We also removed variants with MAF > 4% at the Hadassah in-house database (~1000 ethnic matched exome analyses) or variants that were predicted to be benign (Mutation Taster, http://mutationtaster.org/). After filtering, 33 and 8 homozygous variants remained for patient III-3 and patient V-2, respectively. Only one potentially pathogenic variant, c.961C>T in *TOR1AIP1*, was shared between both

**Table 1 Clinical features of affected individuals**

| Patient | I-2 | I-3 | I-4 | II-1 | III-3 | IV-4 | V-2 |
|---|---|---|---|---|---|---|---|
| Current age (y) | Died at 8.5 | Died at 9 | Died at 5 | Died at 8.5 | 4 | 3.5 | 6 |
| Gender | Female | Female | Female | Female | Female | Female | Male |
| Gestational age (w) | 38 | 36+5 | 37+5 | 38 | 37+2 | 37 | 37 |
| Birth weight (g; centile) | 2400 (1.6%) | 2130 (0.5%) | 2490 (2.7%) | 3300 (33%) | 2310 (1.1%) | 2350 (1.4%) | 2400 (1.6%) |
| HC at birth (cm; centile) | 32 (4.5%) | 31 (3%) | 32.5 (11.5%) | 33.5 (36%) | 30.5 (0.2%) | 32 (5%) | NA |
| Congenital cataract (age of diagnosis) | At birth | Intrauterine | Intrauterine | At birth | Intrauterine | At birth | At birth |
| SNHL (age of diagnosis) | 2 mo | 1 mo | At birth | 2 mo | At birth | At birth | 2 y |
| Cachexia | Severe | Severe | Extreme | Extreme | Extreme | Extreme | Extreme |
| Congenital heart defect | No | TOF | Large VSD, moderate PS, small PDA | Moderate PDA, bicuspid aortic valve | Moderate PDA | Small PDA | TOF |
| Gastrointestinal involvement (age) | No | Ileal atresia (at birth); Gastrointestinal bleeding (4 y) | GER (6 mo) | Chronic constipation (4 y) | GER (2mo) Chronic constipation (2 y) | GER (1 y) | No |
| Renal involvement | No | No | No | Nephrolithiasis | No | Nephrolihtiasis | No |
| Skin manifestations | No | No | No | No | Hypertrichosis | Hypertrichosis | No |
| Gastrostomy (age of insertion) | 8 y | No | No | 4 y | 3 y | 2 y | No |
| PICU admissions | 2 | 2 | No | 4 | 1 | 2 | No |
| Brain MRI (age preformed) | ND | ND | Ponto-cerebellar hypoplasia (prenatal) | CorA, CerA, hypolastic CC, abnormal myelination (1.5 y) | CorA, CerA, hypolastic CC, abnormal myelination (0.5 y, 3 y) | CorA, CerA, hypolastic CC, abnormal myelination (1 y) | ND |

w = weeks, mo = months, y = years, d = days, HC = head circumference, NA = not available, SNHL = sensorineural hearing loss, TOF = tetralogy of Fallot, VSD = ventricular septal defect, PDA = patent ductus arteriosus, PS = pulmonic stenosis, GER = gastro-esophageal reflux, PICU = pediatric intensive care unit, ND = not done, CorA = cortical atrophy, CerA = Cerebellar atrophy, CC = corpus callosum

patients, which were very similar phenotypically. The c.961C>T variant results in a stop codon and a predicted truncation of the protein product at amino acid 321 (Fig. 2a, b). The mutation was verified by Sanger sequencing in both patients, as well as in all affected individuals, and their parents were accordingly heterozygous, as expected. The mutation co-segregated perfectly in the affected state in all the families, confirming it is associated with the disease (Supplementary Fig. 1). We examined the known single-nucleotide polymorphisms (SNPs) around the mutation site in both patients and noted that within the genomic region Hg19 Chr1: 178,063,551–183,920,391 there were 260 consecutive, uninterrupted, homozygous SNPs shared by the two patients. This block of shared haplotype is indicative of a common ancestor, i.e. that the p.Arg321* mutation in TOR1AIP1 is a Palestinian founder mutation.

**Patient fibroblasts lack both major isoforms of LAP1.** Since all the patients were found to be homozygous for the c.961C>T nonsense mutation, they were expected to lack the full-length protein product(s) of the gene and might have been predicted to contain shorter products truncated at amino acid 321, right before the transmembrane segment (see Fig. 2b). To test this, we prepared total cell lysates from primary skin fibroblasts derived from two patients (III-3, IV-4) and three unrelated controls. Proteins were analyzed by immunoblotting, using a polyclonal antibody generated against residues 175–225, which recognizes

two major protein bands in all the controls (Fig. 2c). These bands correspond to the previously reported LAP1B and LAP1C isoforms (with LAP1C arising from an internal translation initiation site at codon 122)[15,18] and both of them are clearly absent in the patient fibroblasts (Fig. 2c, top). Surprisingly, we only observed faint traces of potential truncated forms of the protein in these patient cell lysates (see Supplementary Fig. 2d for a longer exposure of the same blot). We note that a higher molecular weight cross-reactive band, potentially equivalent to mouse LAP1A, has been reported in adult muscle and liver samples from humans[18] but has not been confirmed by subsequent work at the transcript and proteomic level[15]. We did not detect any sign of a higher molecular weight isoform, but note that the unique position of the nonsense mutation, preceding the transmembrane segment, would also be predicted to affect such a product of the TOR1AIP1 gene (Fig. 2b, c).

The protein levels of three major LAP1 interaction partners, lamin A, lamin C and torsinA, as well as the torsin activator LULL1, do not appear to be affected by the loss of LAP1B and LAP1C in patient fibroblasts (Fig. 2c and Supplementary Fig. 2a, c). By contrast, emerin protein levels appeared to be elevated in the two patient fibroblasts in comparison with the controls (Fig. 2c and quantified in three independent lysates as shown at the bottom). We note that a minor torsinA band, potentially corresponding to the proteolytically cleaved form of the protein, appears in the patient cell lysates (TorsinAp in Supplementary

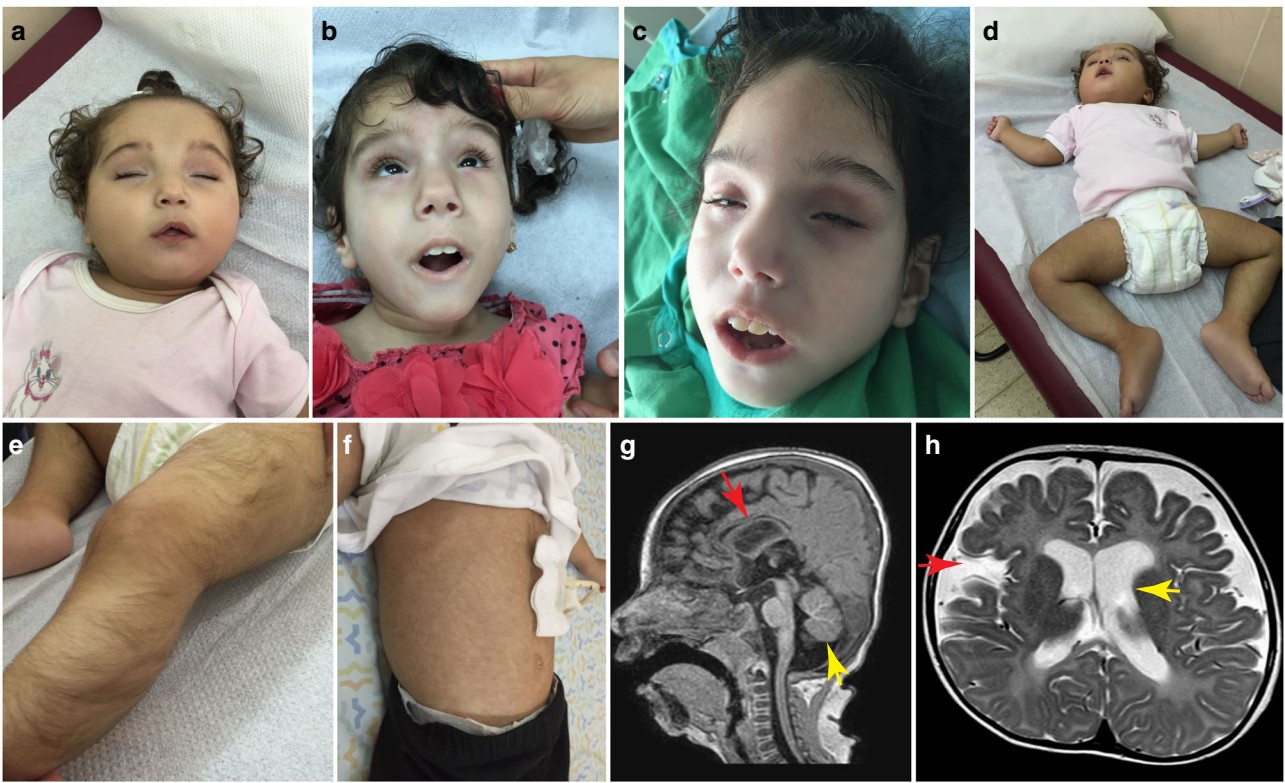

**Fig. 1** Typical clinical features in patients. **a–c** Characteristic facial features including cataract extraction (**b**), microphthalmia, deep set eyes, and tented upper lip in two affected individuals (III-3, IV-4). Patient III-3 is shown at age of 1.5 years (**b**) and 3 years (**c**). **d** Severe truncal hypotonia featured by "frog-like" lying position. **e** Lower limb hypertrichosis. **f** Gastrostomy tube as well as vascular skin changes (livedo reticularis). **g** T1-sagital MR in patient III-3 showing hypoplastic corpus callosum (red arrow), cerebellar atrophy (yellow arrow), and cortical atrophy. **h** T2-axial magnetic resonance in patient IV-4 showing global cortical atrophy as manifested by enlarged ventricles (yellow arrow) and wide open Sylvian fissures consistent with temporal atrophy (red arrow)

Fig. 2a). This proteolytic cleavage has previously been shown to occur in response to ER stress and other stimuli[26].

Our results suggest that the main effects of the mutation can be traced to a loss-of-function phenotype resulting from the absence of the two major LAP1 protein isoforms in patient cells.

**Patient cell LAP1 transcript levels are strongly reduced**. To address the potential regulation of *TOR1AIP1* and related gene products at the mRNA level, we used quantitative real-time polymerase chain reaction (qPCR). Two patient-derived primary fibroblast lines were compared to two controls in three independent experiments. As shown in Fig. 2d, LAP1 mRNA levels were strongly reduced in both of the patient-derived fibroblast cultures and represent <40% of the controls (see also Supplementary Fig. 2b for a second set of qPCR primers). Emerin mRNA levels were only slightly elevated in patient-derived fibroblasts. However, when the results of both qPCR and western blotting (WB) for emerin (Fig. 2c, d) are considered, there appears to be trend toward a compensatory increase in emerin in patient fibroblasts (see Supplementary Fig. 3b for immunostaining of emerin at the NE). Taken together, our results suggest that the main effect of the nonsense mutation in *TOR1AIP1* is a reduction in the level of mature transcripts available for the translation of LAP1 protein isoforms, with the possible added effect of rapid degradation of short translation products (see Supplementary Fig. 2d).

**Abnormal nuclear phenotypes are observed in patient cells**. We next asked whether the absence of both LAP1 isoforms affected the organization of the nuclear lamina and adjacent cellular

structures in patient-derived fibroblasts. We used indirect immunofluorescence (IF) with primary antibodies directed against LAP1, emerin, and lamin A/C to compare fibroblasts from the two available patients (III-3, IV-4) to control fibroblasts from an unaffected individual.

Fibroblast nuclei are extremely flat (see also: confocal and electron microscopic images, below) and typically appear by immunostaining with NE markers as a nuclear disk with pronounced rim staining (Fig. 3a and Supplementary Fig. 3). As expected, widefield IF with anti-LAP1 antibodies showed this NE staining pattern only for control fibroblasts (Supplementary Fig. 3a). A pronounced nuclear rim was clearly evident by anti-emerin staining and its overall staining pattern appeared similar in the control and patient-derived fibroblast lines (Supplementary Fig. 3b). By contrast, anti-lamin nuclear rim staining appeared diminished in many of the patient-derived fibroblast nuclei as determined by confocal microscopy with double staining for lamin A/C and the ER marker protein disulfide isomerase (PDI; Fig. 3a). The position of the NE could be identified at the border of anti-PDI staining in both control and patient-derived fibroblasts and the quantification of anti-lamin staining intensity at the nuclear rim showed a reduction to 70.8% and 66.5% relative to control fibroblasts (Fig. 3a, bar chart on the right; see Methods for further details of the quantification of nuclear rim staining).

In addition to this change in anti-lamin staining, we noticed that some of the patient-derived nuclei were somewhat distorted in shape and contained holes suspected to be cytoplasmic channels traversing the nucleus. These putative channels were observed both by Hoechst 33258 staining of the DNA and by anti-lamin staining (Fig. 3b, arrowheads), but were completely

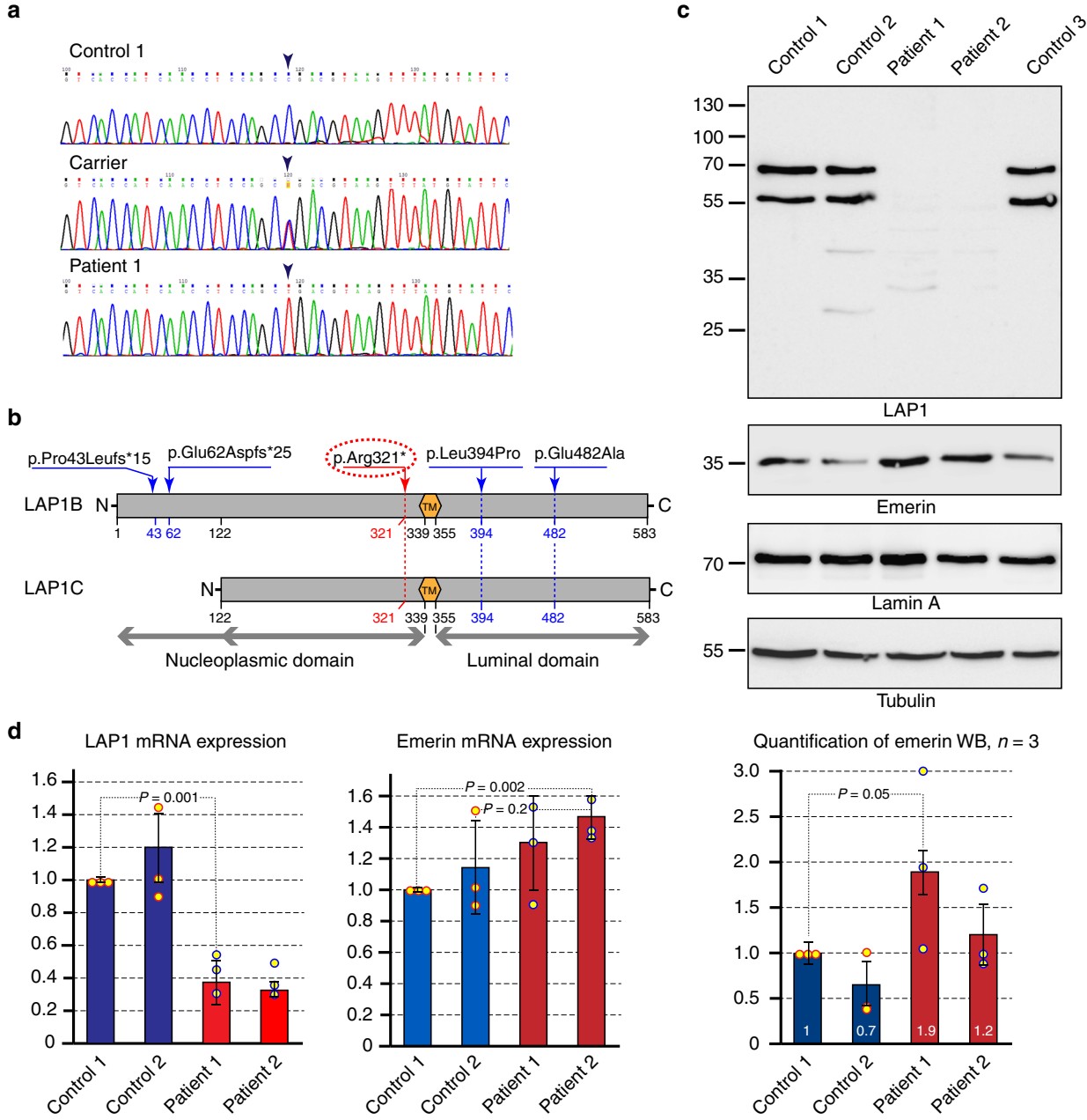

**Fig. 2** A nonsense mutation in *TOR1AIP1* results in the loss of both lamina-associated polypeptide 1 (LAP1) isoforms. **a** Sanger sequencing of the *TOR1AIP1* c.961C>T variant in genomic DNA of wild-type, heterozygous, and homozygous individuals. The variant position is marked by arrowheads. **b** Schematic representation of the major LAP1 protein isoforms and previously described mutations[23-25]. TM transmembrane segment. The shorter LAP1C isoform is thought to arise from internal translation initiation at codon 122. Two previously described frameshift mutations are only predicted to affect the LAP1B isoform and result in short truncation products, while two missense mutations are situated in the luminal domain. The nonsense mutation in codon 321 identified in the current study is highlighted in red. See ref. [42] for further details. **c** Western immunoblot analysis of cell lysates from cultured primary skin fibroblasts of two patients (Patient 1: III-3; Patient 2: IV-4) and three unrelated controls. Protein blots were reacted with antibodies directed against: LAP1, emerin, lamin A, and α-tubulin. The polyclonal anti-LAP1 antibody was generated against a region that is common to both LAP1B and LAP1C isoforms (as seen in the controls) and would thus be expected to recognize truncated N-terminal fragments. See Supplementary Figure 2d for a longer exposure of this blot. A quantification of emerin band intensity in three independent lysates and immunoblots is shown at the bottom (see: Methods; bars indicate SEM). **d** Mature mRNA levels were analyzed by quantitative real-time polymerase chain reaction for primary skin fibroblasts derived from two patients and two controls. The results for LAP1 and emerin were normalized to β-tubulin and represent the means and s.d. of three independent experiments

absent from control fibroblast lines. The suspected channels were highly irregular in shape and their apparent dimensions ranged from 0.2 to 5.3 μm with a mean of 1.4 μm in the short axis to 0.3–13.2 μm with a mean of 2.7 μm in the long axis (measured by light microscopy in 47 patient cell nuclei from both patients). They occurred in 10.1% of the nuclei of Patient 1 (III-3) and 6.3%

of the nuclei of Patient 2 (IV-4) fibroblasts, with some nuclei containing 2–3 suspected channels (Fig. 3b; bar chart and gallery at the bottom). We note that DNA staining alone may not be sufficient for the identification of these structures because of the presence of nucleoli, making double staining with NE markers the preferred choice.

We next used confocal microscopy and monoclonal antibody mAb414 to stain nuclear pore complexes in the NE of control and patient-derived cells. The staining pattern of the flat fibroblast nuclei is very similar in the control and patient nuclei, with a single optical section through a patient nucleus showing a suspected channel in cross-section (Fig. 4, middle). A z-stack of optical sections shows that this channel can be followed through the nucleus from top-to-bottom and would thus be expected to open into the cytoplasm on both ends and form a genuine nuclear-spanning channel (Fig. 4, right).

**Electron microscopy reveals large nuclear-spanning channels.** To gain a better understanding of the cytoplasmic channels observed by light microscopy, we employed transmission electron microscopy and prepared samples in two different ways to obtain side and top views of the flat fibroblast cells (see: Methods). When viewed from the side, the average height (z axis) of fibroblast nuclei was about 1 micron (Fig. 5a, b, top panels). The top panels in Fig. 5b show a side view of a patient-derived fibroblast nucleus transected by a narrow cytoplasmic channel, demarcated on both sides by a double membraned NE. Top views of fibroblasts were obtained by fixation and embedding of attached cells directly on coverslips (see: Methods for further details), resulting in ultra-thin sections that were parallel to the coverslip plain. These top views produced typical images of disk-shaped flat nuclei in control and patient-derived cells (Fig. 5a–c) and revealed that the cytoplasmic channels in patient-derived nuclei are often quite extensive and irregularly shaped (Fig. 5b, middle panels). The channels contain trapped cytoplasmic organelles like mitochondria and rough ER vesicles and are fully encompassed by an intact NE containing nuclear pore complexes (Fig. 5b, bottom panels).

Higher magnifications of the NE and a comparison of control and patient-derived nuclei show no evidence for herniation or the protrusion of nuclear contents through openings in the membranes. The distance between the two nuclear membranes appears to be normal in patient cell nuclei, both in the nuclear periphery and around the cytoplasmic channels (Fig. 5a–c, bottom panels). One more example of the images obtained in top views of patient-derived fibroblast nuclei is shown in Fig. 5c, with several instances of smaller suspected channels seen in cross-section and clearly encompassed by double membranes. These structures would be too small for identification by light microscopy, suggesting that the deduced occurrence of channels in patient cells (Fig. 3b) may be an underestimate. However, since these smaller structures were only viewed in single ultra-thin sections, we cannot exclude the possibility that they represent invaginations of the NE rather than complete channels that span the height of the nucleus.

Taken together, the different imaging techniques demonstrate that large cytoplasmic channels may constitute a considerable disruption of nuclear architecture in patient-derived fibroblasts. Similar cytoplasmic channels have previously been reported in the nuclei of human and mouse cells treated with farnesyltransferase inhibitors and were attributed to a centrosome separation defect and to the perturbed functions of lamin B1 and pericentrin[27]. Our observations suggest that the absence of the two major isoforms of the LAP1 protein may be affecting the complex network of interactions extending from the inner nuclear membrane to both sides of the NE.

**Single-cell phenotypes differ from laminopathy hallmarks.** Severe changes in nuclear morphology have been described for various laminopathies, although these phenotypes do not necessarily differentiate one disease from another[28,29]. As mentioned above, we observed no evidence for herniation or a damaged NE

in patient-derived fibroblasts. A comparison of nuclear morphology parameters measured by light microscopy in control and patient-derived fibroblasts showed no significant difference in the calculated nuclear volume or circularity, while small but significant changes were measured for nuclear height and area (Supplementary Fig. 4). The lower height and larger area measured in patient nuclei compared to the controls suggest that the absence of the LAP1 isoforms may be affecting the rigidity of the NE in fibroblasts. The distribution of nuclear pore complexes in the NE of patient-derived fibroblasts remained normal, as determined by confocal microscopy and by direct surface imaging of the NE using scanning electron microscopy (Supplementary Fig. 5).

Other well-known cellular effects observed in different laminopathies include slower proliferation, increased DNA damage and apoptosis, and premature cellular senescence[28,30–32]. Primary fibroblast cell lines from two patients (III-3, IV-4) and two unrelated controls were grown under identical conditions and tested for these features. As shown in Supplementary Fig. 6, patient fibroblasts do not lag behind the controls and in fact both of the patient-derived cell lines show a slightly increased proliferation rate relative to controls. Staining with fluorescent markers for apoptotic and necrotic cell death, as well as anti-cleaved caspase-3, showed very low staining levels and no difference between control and patient-derived fibroblasts (Supplementary Fig. 7a, b). We note that, for both patients, the small number of cells that stained for apoptosis/necrosis markers did not correspond to those containing cytoplasmic channels in their nuclei. Cellular senescence was also observed at a lower frequency in patient versus control cell lines (~13% versus ~21%; Supplementary Fig. 7c). To test for potential perturbations in DNA damage response pathways, we assayed fibroblasts for the presence of two known markers: the phosphorylated form of histone H2AX (γH2AX) and the p53-binding protein 1 (53BP1). As shown in Supplementary Fig. 8, both markers can be observed in a small number of foci in most of the nuclei of patient-derived fibroblasts, without prior exposure to DNA-damaging agents. This may only be considered to be a mild defect in the regulation of the DNA damage response, but it likely represents an indirect effect mediated through changes in the organization of the lamina.

Overall, at the single-cell level, patient-derived primary fibroblasts homozygous for the LAP1 nonsense mutation exhibit a set of phenotypes that are distinct from those reported for HGPS and other severe laminopathies.

**Impaired cellular motility in patient-derived fibroblasts.** One potential consequence of disrupting the interactions between the NE and the cytoskeleton may be impaired cell motility[33,34]. Indeed, Nery et al. have previously reported that torsinA knockout mouse embryonic fibroblasts showed delayed directional migration[22]. Moreover, both torsinA and LAP1 have recently been shown to be indispensable for rearward nuclear movement during centrosome orientation in migrating fibroblasts[35].

To measure cell migration in vitro and compare the different primary skin fibroblast lines, we used the well-established wound healing scratch assay. An automated set-up allowed us to quantify the cell migration rate of patient versus control fibroblasts using time-lapse microscopy and multiple replicates. As shown in Fig. 6a, the fibroblasts from both available patients lagged significantly behind control fibroblasts in their wound closing ability, tested on a 10 μm scratch created in a cell monolayer. At the 52-h time point, control fibroblasts had largely closed the gap and achieved well over 80% confluence in the scratch wound, while patient-derived fibroblasts had failed to fill large gaps in the wound area (Fig. 6a, left). Plotting wound confluence over time in

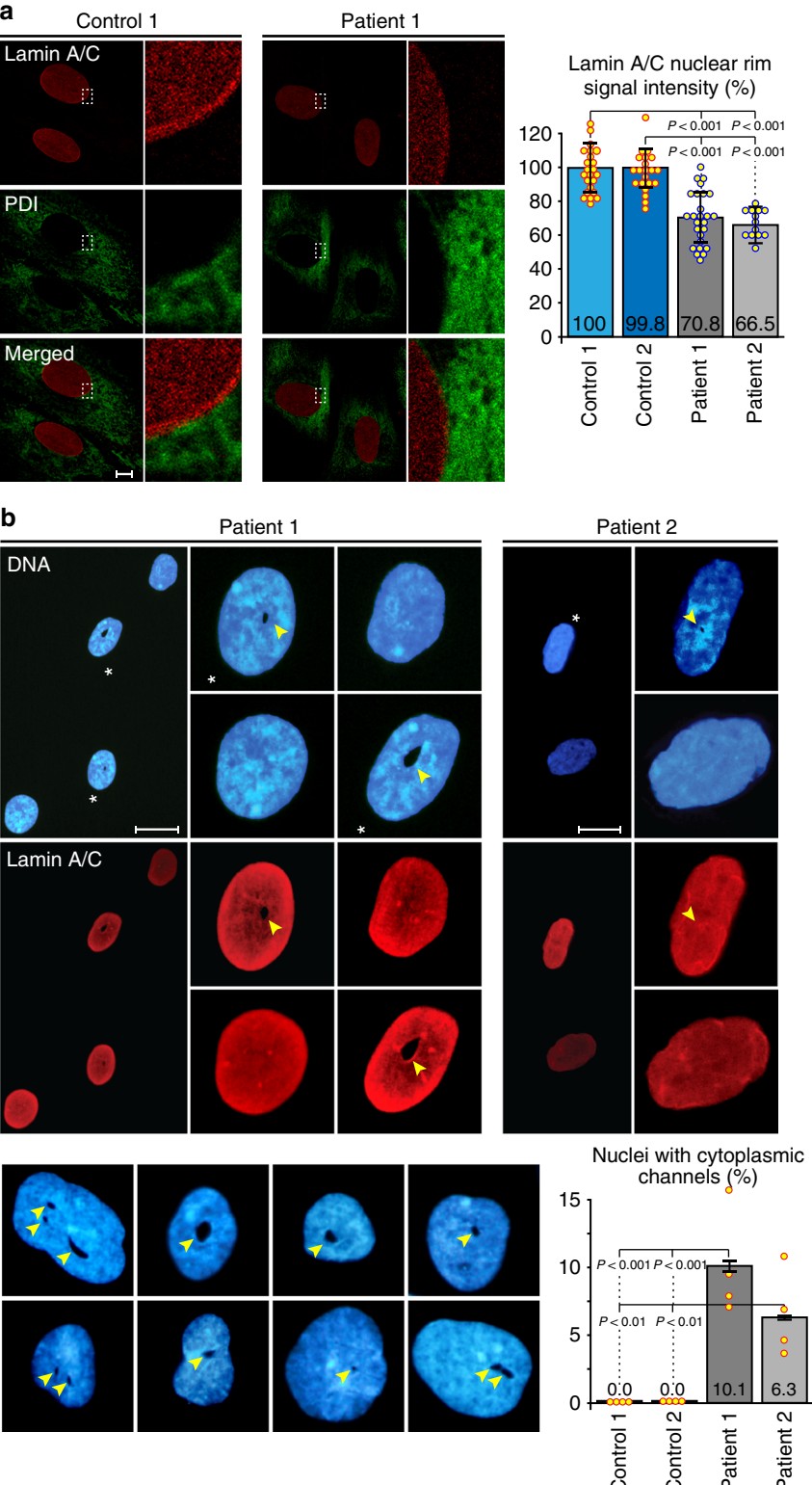

six replicates for each primary cell line demonstrated that normal fibroblasts closed the scratch wound earlier than patient-derived fibroblasts, which in fact failed to reach complete confluence in over 5 days (Fig. 6a, right; see also: Supplementary Movies 1–3).

We next tested the migration properties of sparsely plated control and patient-derived fibroblasts outside the context of a wound. Trajectory plots of individual cells and the quantification of several motility parameters are shown in Fig. 6b. The results reveal a substantially reduced motility in patient-derived fibroblasts under these two-dimensional (2D) random migration conditions. Importantly, individual patient cells not only exhibit reduced velocity but also a shorter Euclidean distance (reflecting the distance between the starting and end points of migration) and the fact that these cells explore a much smaller area.

Taken together, our results demonstrate a severe effect of the *TOR1AIP1* nonsense mutation on the normal cellular motility

**Fig. 3** Abnormal nuclear phenotypes are evident in patient-derived fibroblasts. **a** Control and patient-derived (Patient 1: III-3; Patient 2: IV-4) fibroblasts were immunostained with anti-lamin A/C and anti-protein disulfide isomerase (anti-PDI) antibodies and analyzed by confocal microscopy. Representative images are shown on the left and selected areas are enlarged. The edge of anti-PDI staining marks the nuclear envelope (NE) and a pronounced anti-lamin nuclear rim staining can be identified in the control. Quantitative analysis of anti-lamin staining intensity was performed for 2 control and 2 patient cell lines with a custom written ImageJ macro defining a 1 μm wide ring around the NE (see: Methods for further details). A summary of the relative staining intensity measurements is shown on the right ($n \geq 30$; bars indicate SEM). Scale bar, 1 μm. **b** Cytoplasmic channels observed by DNA (Hoechst 33258) and anti-lamin A/C staining, using widefield immunofluorescence, in some of the nuclei from both patient-derived primary fibroblast cell lines. Asterisks indicate nuclei containing channels, shown in a wide field view and at a higher magnification. Channels are marked by arrowheads. The gallery at the bottom shows a collection of DNA-stained nuclei with distorted shapes containing 1–3 channels of different sizes. Such channels were only observed in patient-derived fibroblast nuclei. Quantitative analysis of channel occurrence in 2 control and 2 patient cell lines is shown in the bar chart (bottom-right; $n = 150$ in 4 independent experiments; bars indicate SEM). Scale bars, 20 μm

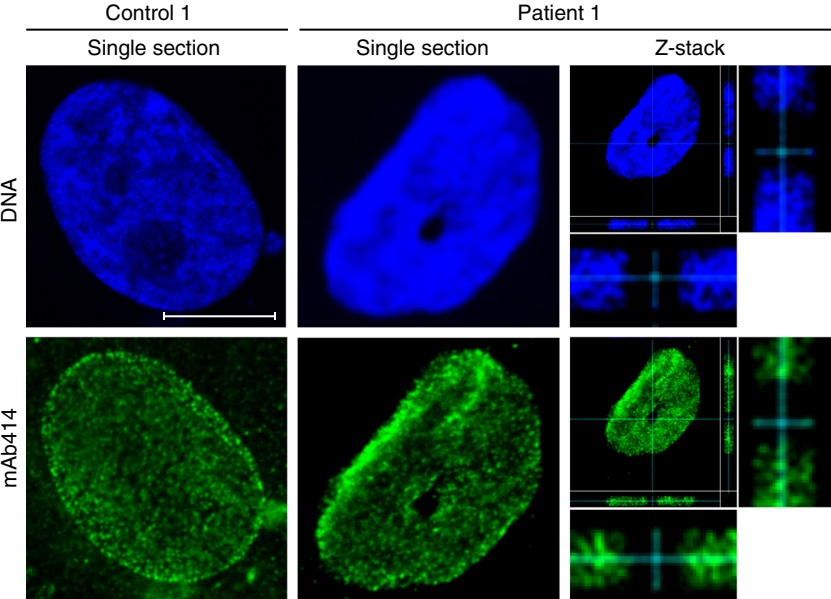

**Fig. 4** Cytoplasmic channels cross patient cell nuclei from top to bottom. Nuclear pore complexes in the nuclear envelope (NE) of control and patient-derived nuclei were stained by indirect immunofluorescence using mAb414 and imaged by confocal microscopy. Single optical sections are shown for one control and one patient-derived nucleus. A full z-stack of 27 optical sections, taken with a 70 nm step size through the patient-derived fibroblast, is shown on the right. Note the enlarged side views of the x and y axes demonstrating that the channel cuts across the whole nucleus. See Supplementary Fig. 5 for an assessment of nuclear pore complex distribution in the NE. Scale bar, 10 μm

function of skin fibroblasts. They may also provide a conceptual framework for understanding additional consequences in other cell types (especially neuronal cells), explaining the multisystemic clinical manifestations of the disease.

**LAP1-coding constructs rescue multiple cellular phenotypes.** To ask whether the phenotypes observed at the single-cell level in patient-derived fibroblasts were directly related to the absence of the LAP1 isoforms, we attempted to rescue these defects by re-expressing either LAP1B or LAP1C in these cells. Primary fibroblasts from Patient 1 (III-3) were transduced with lentiviral particles to generate stably transfected patient cell lines expressing LAP1B or LAP1C coding sequences under the EF1a promoter or lines stably transfected with the equivalent empty vectors as negative controls. Control fibroblasts were also transfected with the empty vectors to provide positive controls (see: Methods for further details). As shown in Fig. 7a, both the LAP1B construct and the LAP1C construct caused a strong reduction in the occurrence of cytoplasmic channels in Patient 1 cells as compared to the empty vector controls. Patient 1 cells expressing the empty vector markers exhibited ~10% of nuclei containing channels (compare: Fig. 3b), while expression of the LAP1B and LAP1C

constructs reduced this to ~3% and ~4%, respectively (Fig. 7a). Collective cell motility, as determined in the wound healing scratch assay, was similarly affected by both of the LAP1B and LAP1C constructs providing a considerable increase in the wound closing ability of patient-derived cells compared to the equivalent empty vector controls (Fig. 7b). Interestingly, when random migration of individual sparsely plated cells was tested the degree of rescue obtained by stable transfection of LAP1B was substantially higher than that of LAP1C. This can be seen in the individual trajectory plots (Fig. 7c, left), as well as in the quantified parameters: LAP1B rescued velocity and accumulated distance from 58% to 82% of the values measured in the positive control and rescued Euclidean distance from 38% to 69%. By contrast, LAP1C only rescued velocity and accumulated distance from 55% to 70% and had no significant effect on the Euclidean distance (Fig. 7c, right). Supplementary Fig. 9 shows a potential explanation for some of these differences obtained by immunoblot analysis of total cell lysates from the different fibroblast cell lines. The results suggest that stably transfected LAP1B is expressed in patient cells at a similar level to the normal isoform in control fibroblasts, while the LAP1C isoform is only detected at a low level in transfected patient cells. Figure 7d shows an attempt

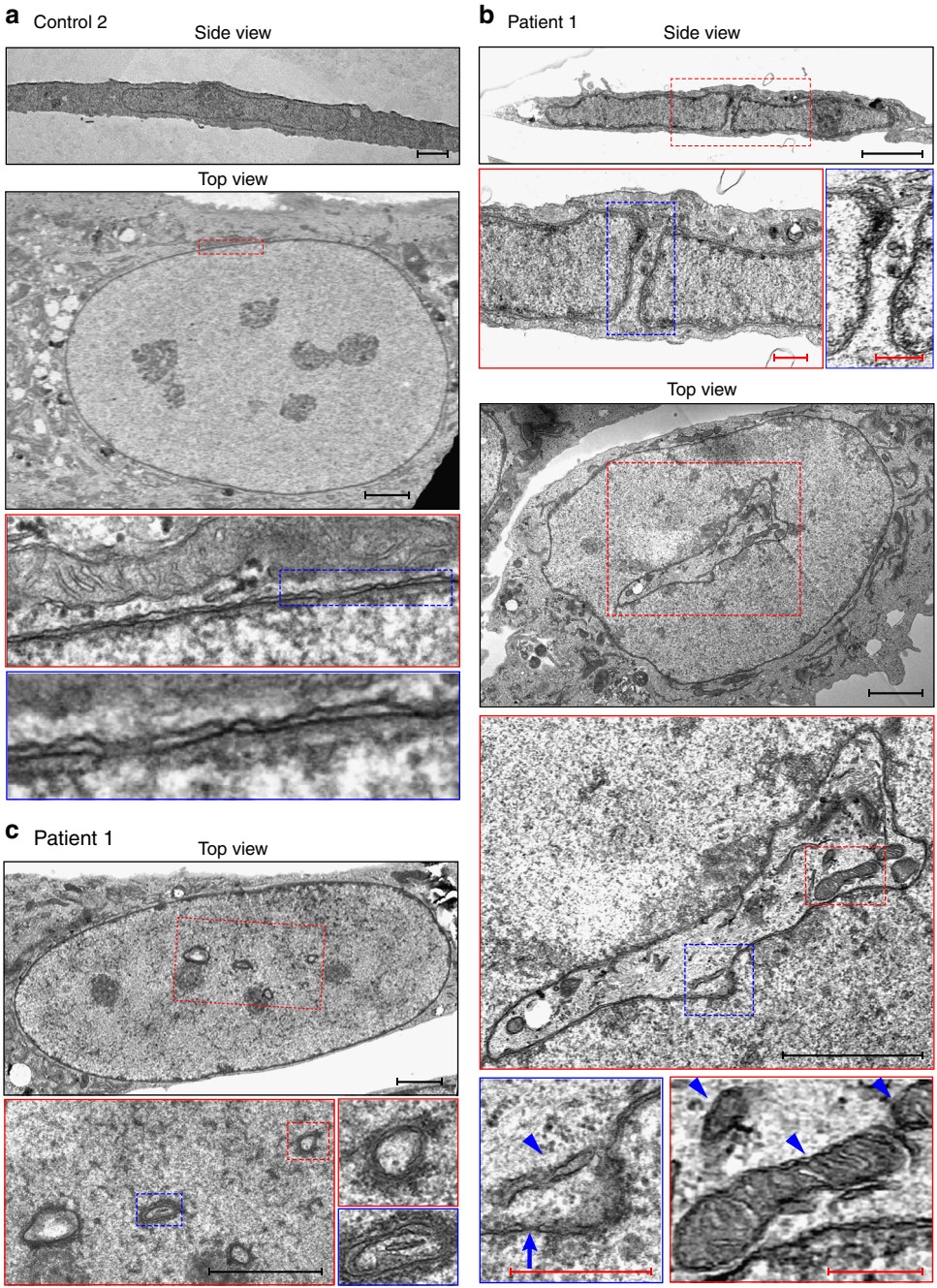

**Fig. 5** Patient cell nuclei contain intricate channels with trapped cytoplasmic organelles. Control (**a**) and patient-derived primary fibroblasts (**b**, **c**) were prepared for transmission electron microscopy as described in Methods. Dashed red and blue lines indicate areas enlarged in subsequent panels. Side views were obtained by the fixation of attached cells followed by scraping off the surface and further fixation, embedding, and preparation of thin sections. Top views were obtained by a different preparation protocol, in which cells were fixed and embedded directly on a coverslip surface, to ensure an identical flat orientation of all the cells. **a** Side and top views of control fibroblasts exhibiting normal nuclear morphology and no cytoplasmic channels. Enlarged areas show the two nuclear membranes, the perinuclear space, and nuclear pore complexes in cross-section. **b** Side and top views of patient-derived fibroblasts. In the side view, a cytoplasmic channel transects the flat nucleus from top to bottom. Note cytoplasmic organelles near the top opening of the channel. The top view shows a single cell containing an intricate, irregularly shaped channel, which is fully encompassed by a double membraned nuclear envelope. Double membranes and a typical nuclear pore in cross-section (arrow) can be identified in the bottom left panel. Arrowheads point to trapped cytoplasmic organelles. **c** Additional top view images from a different patient-derived fibroblast. Note several smaller suspected channels, seen in cross-section and clearly encompassed by double nuclear membranes. Scale bars: black 2 µm, red 400 nm

to achieve combined rescue with both the LAP1B and LAP1C constructs, as compared to a control with combined empty plasmids. The random migration of sparsely plated cells was followed in this experiment, since the expression of both green fluorescent protein (GFP) and mCherry could be visually verified

in the individually tracked cells. Although the combined expression rescued the impaired motility in this assay, it did not produce a clear synergistic effect in comparison to the expression of LAP1B alone (Fig. 7d).

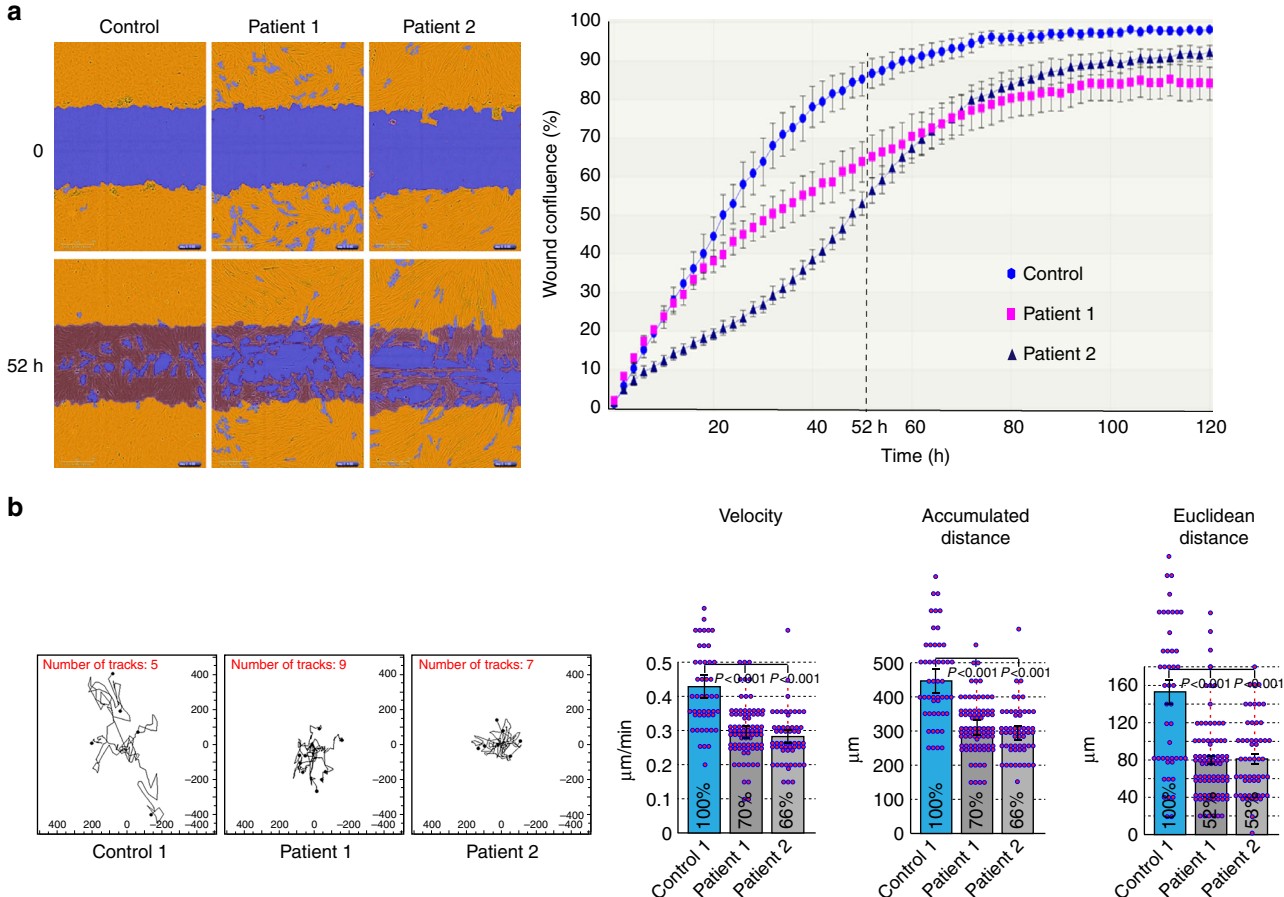

**Fig. 6** Impaired cellular motility in patient-derived fibroblasts. **a** Quantitative analysis of the wound-closing activity of control and patient-derived fibroblasts in the IncuCyte ZOOM™ scratch wound assay. Cells were grown in 96-well plates to form a monolayer and uniform scratch wounds were made with WoundMaker™. Right: wound confluence graphed over time for one control and two patient-derived cell lines recorded every 2 h for a total duration of 126 h, showing the means ± SEM for six replicates in each cell line. Left: segmented and masked representative images showing the initial position ($t = 0$) and the extent of wound closure after 52 h. A significant delay in cell migration and wound closing was observed in both of the patient-derived lines. **b** Random two-dimensional (2D) cell motility of sparsely plated control and patient-derived fibroblasts. Left: trajectory plots of individual cells from one control and two patient-derived cell lines. Images were acquired every 30 min for a total of 20 h using the IncuCyte ZOOM™ acquisition software. Right: quantitative analysis of the velocity, accumulated distance (total cell path length), and Euclidean distance (shortest distance between starting and end point of migration); $n = 49$ (Control), $n = 83$ (Patient 1), $n = 56$ (Patient 2); bars indicate SEM. Reduced 2D motility is observed for both patient lines in all the parameters

In summary, re-expression of LAP1B or LAP1C in patient-derived fibroblasts rescued three distinct cellular phenotypes and hinted at a differential effect of the two isoforms, especially in the case of the Euclidean distance measured in the trajectory plots of individual fibroblasts.

## Discussion

The present study describes seven patients manifesting a distinct phenotype dominated by early progressive neurological impairment (profound global psychomotor retardation, severe microcephaly, and global brain atrophy as evidenced by MRI), congenital heart malformations, congenital cataract, and moderate intra-uterine and severe post-natal growth retardation. The disease is associated with homozygosity for a *TOR1AIP1* nonsense mutation that predicts an absence of the mature protein due to premature truncation. Our WB results indeed demonstrate the absence of both the LAP1B and LAP1C isoforms in patient-derived fibroblasts that are not replaced by stable shorter translation products. This is likely explained by a substantial decrease in mature mRNA levels, as observed by qPCR, and suggests the potential involvement of rapid cytoplasmic RNA surveillance

mechanisms such as nonsense-mediated mRNA decay[36]. Protein turnover may also contribute to the reduction in LAP1 levels, since faint lower molecular weight bands are observed by immunoblotting of patient cell lysates (Supplementary Fig. 2d). We note that the apparent compensatory increase we observed in emerin transcript and protein levels in patient cells fits the synthetic lethality and compensatory interactions between emerin and LAP1 reported in mice[18].

Although primary skin fibroblasts differ from the main types of cells and tissues affected by neurogenetic diseases, they are often the only available source from patients. We asked whether we could identify distinct phenotypes in patient-derived fibroblasts at the single-cell and single-nucleus level. Such changes, hinting at perturbed functions of the LAP1 protein, were observed in anti-lamin A/C nuclear rim staining, in the presence of cytoplasmic channels transecting the nucleus and in a severe reduction in cellular motility. These findings suggest that a true loss-of-function mutation in LAP1 has general consequences in many cell types, although they do not exclude the possibility that specific cell types, such as neurons, eye lenses, or muscles, may be more severely affected. Members of the LINC complex bind

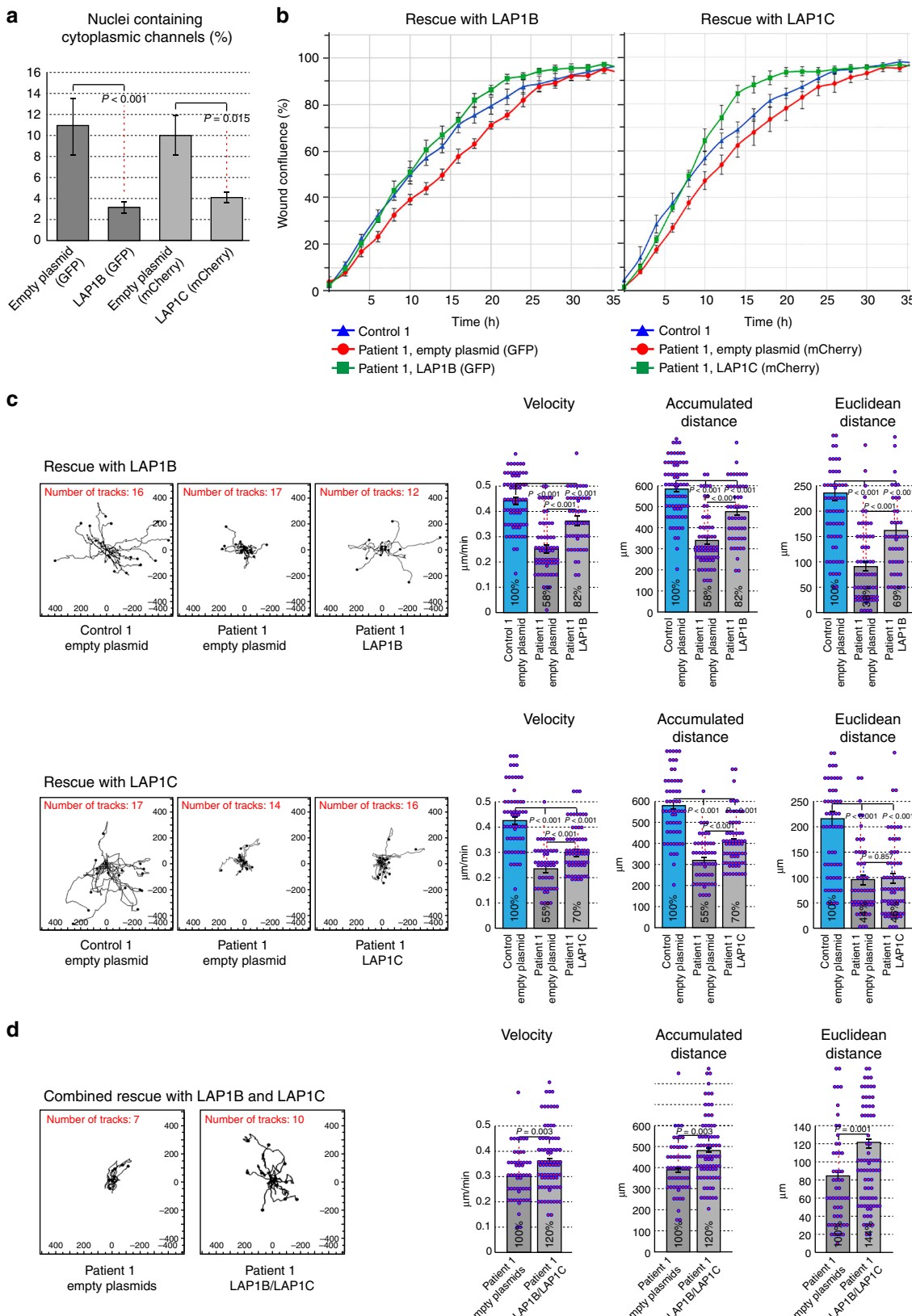

within the perinuclear space and their interactions extend to both sides of the NE[7,37]. LAP1 binds and activates the ATPase activity of torsinA, while torsinA has been shown to bind the KASH domains of LINC complex components[21,22]. Thus, simply by virtue of its topology and interaction partners, LAP1 is a classical candidate for being a NE toolbox protein required for nuclear

positioning and cell motility[33,38–40]. More specifically, both torsinA and LAP1 are required for the polarization of fibroblasts for migration, which includes a rearward movement of the nucleus and the establishment of anterior centrosome orientation[35]. Our results support a direct role of LAP1 in the motility of fibroblasts, demonstrated by profound defects in collective cell movement in

**Fig. 7** Rescue of nuclear morphology and cell motility defects by transduction with lamina-associated polypeptide 1B (LAP1B) and LAP1C coding constructs. Multicistronic lentiviral vectors encoding LAP1B or LAP1C driven by the EF1a promoter and containing two internal ribosome entry site sequences, EGFP or mCherry, and a puromycin resistance marker were constructed and lentiviral particles were prepared as described in Methods. Control or Patient 1 primary fibroblast lines were transduced with lentiviral particles containing LAP1B-coding, LAP1C-coding, or empty plasmid controls. Following several rounds of antibiotic selection and visual verification of fluorescent marker expression in all cells, the stably transfected cell lines were analyzed in three separate assays all demonstrating partial but significant reversal of the defects. **a** The occurrence of cytoplasmic channels in the nuclei of stably transfected Patient 1 cell lines was analyzed by widefield immunofluorescence, as in Fig. 3b ($n = 110$ in 2 independent experiments, performed in duplicates; bars indicate SEM). **b** Wound closing activity was analyzed simultaneously for three cell lines at a time in the scratch wound assay, as in Fig. 6a. Original untransfected control fibroblasts were compared to Patient 1 cell lines stably transfected with the LAP1B-coding or LAP1C-coding and the equivalent empty plasmid controls. Wound confluence was graphed over time with images recorded every 2 h for a total duration of 72 h, showing the means ± SEM for six replicates in each cell line. **c** Random two-dimensional (2D) cell motility was analyzed by trajectory plots and quantitative analysis, as in Fig. 6b. $n = 73$ (Control, empty plasmid), $n = 62$ (Patient 1, empty plasmid), $n = 52$ (Patient 1, LAP1B); $n = 58$ (Control, empty plasmid), $n = 51$ (Patient 1, empty plasmid), $n = 67$ (Patient 1, LAP1C); bars indicate SEM. **d** An attempt to achieve combined rescue with both LAP1B and LAP1C coding constructs. A direct comparison is shown between Patient 1 cells stably transfected with the two empty plasmids and with the two LAP1-coding plasmids. Note that this experiment cannot be conducted under optimal conditions because of the lack of a second selection marker, although GFP and mCherry were both observed in all the cells analyzed by trajectory plots. Random cell motility in 2D was analyzed as above. Although rescued motility is observed, the quantitative analysis parameters do not show a synergistic effect beyond what is achieved by the LAP1B construct alone (**c**, top row). $n = 53$ (Patient 1, empty plasmids), $n = 78$ (Patient 1, LAP1B+LAP1C); bars indicate SEM

the wound healing assay, as well as in 2D random migration of individual cells. Importantly, the absence of LAP1 isoforms cannot be compensated by other members of its interaction network in this loss-of-function condition in patient cells.

Another indication for the disruption of normal NE–cytoskeleton interactions is the presence of large cytoplasmic channels in some of the patient cell nuclei. A similar phenomenon, dubbed "donut-shaped nuclei", has been observed following treatment with farnesyltransferase inhibitors and a major drug target was identified in the nuclear lamina[27]. Dynamic tubular channels and shorter NE invaginations have been observed in the nuclei of many cultured vertebrate cell types and have been suggested to correlate with de-differentiation[41]. The dynamics and potential consequences of cytoplasmic channels on nuclear functions in our patient cells have yet to be investigated. However, these channels represent a clear disease-associated cellular phenotype and are not observed at all in control fibroblasts.

At the single-cell level, patient-derived fibroblasts do not exhibit the type of changes in proliferation, apoptosis, and cellular senescence that have been reported for other severe laminopathies[28–30]. Likewise, gross defects in nuclear morphology such as NE herniation, nuclear lobulation, or clustering of nuclear pore complexes are not observed. Thus the homozygous p.Arg321* mutation in *TOR1AIP1* appears to result in a unique set of phenotypes, both at the clinical and single-cell levels.

The previously reported mutation p.Glu62Aspfs*25 (Fig. 2b), which leads to early protein truncation at amino acid 83, results in a unique juvenile onset limb girdle muscular dystrophy associated with cardiomyopathy but sparing central nervous system involvement. The authors showed that the LAP1B isoform was absent upon immunoblotting in skeletal muscle extracts, but an additional protein band around 50 kDa was present in these blots[24]. This additional band was later suggested to correspond to the LAP1C isoform, produced by alternative translation initiation at position 122, and proposed to explain the escape from more deleterious effects of this mutation[42]. A similar report described two siblings with compound heterozygous mutations in *TOR1-AIP1* presenting generalized skeletal myopathy but more severe heart involvement that required heart transplantation due to cardiac failure[23]. Considering these separate reports, although LAP1B is ubiquitously expressed, its function is presumably essential for striated and cardiac muscles. This assumption is further supported by observations from conditional knockout studies of LAP1 in mouse muscles[18,43].

The second human LAP1 isoform LAP1C was identified and shown to be expressed in most tissues, including the brain.

Notably, its expression is cellular differentiation-dependent since it is more abundant in undifferentiated human cell lines compared with LAP1B, but this ratio tends to be reversed with increased maturation[15]. Our study presents the only report of a mutation that abolishes the expression of both of the major LAP1 isoforms in humans. Based on the clinical features and what is known about the expression patterns of the human isoforms, it may be speculated that the exceptionally severe early-onset phenotype experienced in our patients arises from a deficiency of LAP1C rather than of LAP1B. According to this assumption, the lack of myopathy and cardiomyopathy in our patients may be explained by their early lethality, preventing the later occurrence of LAP1B-deficiency effects.

Another homozygous *TOR1AIP1* mutation p.Glu482Ala (Fig. 2b) resulting in a single amino acid change was reported in a patient with childhood-onset progressive dystonia and cerebellar atrophy with a later development of dilated cardiomyopathy[25]. In vitro studies displayed a dual effect of the mutation with mislocalization of LAP1 and its aggregation in the ER as well as decreased amount of the two LAP1 isoforms. Nevertheless, LAP1C levels appear more preserved compared with LAP1B[25]. The later onset of the neurological phenotype and its attenuated severity in this patient compared with our patients may be attributed to the reduced, although not absent, LAP1C.

Collectively, in agreement with the effect of the various reported mutations on both LAP1 isoforms, our study underscores the critical role of LAP1C in fetal and infantile brain development, as well as fetal organogenesis, mainly of the heart and eye lens. The comparison of human laminopathies to mouse disease models is a difficult task, which is compounded in the case of LAP1 by the need to compare different protein isoforms. Despite this caveat, the severe phenotype seen in our patients is somewhat consistent with the observation that Tor1aip1 knockout mice exhibit perinatal lethality and all three isoforms of LAP1 are absent from whole-brain lysates of these mice[44]. Moreover, the observation that the loss of the major human LAP1 isoforms impairs cellular motility provides another potential explanation for the effects on early organogenesis (congenital heart defects, brain development), which depends on collective cell migration and the colonization of specific developmental niches[45–47].

In conclusion, various *TOR1AIP1* mutations may result in a highly variable phenotype depending on the effect of the mutation on the two human LAP1 isoforms and the delicate interplay with interacting proteins in and around the NE. Complete LAP1B and LAP1C deficiency is associated with the most severe phenotype indicating the pivotal role of these isoforms in early life,

mainly during growth and development. By contrast, isolated LAP1B deficiency is associated with late-onset skeletal and cardiac muscle presentation. Intermediate phenotypes may occur depending on the LAP1C/LAP1B ratio and availability in the NE.

## Methods

**Patients**. The study included seven patients from five separate families of Arab Muslim ancestry. All subjects (or legal guardians) gave written informed consent for participation and the study was approved by the Emek Medical Center and Hadassah Medical Center Ethical review boards. The authors affirm that human research participants provided informed consent for publication of the images in Fig. 1.

**Whole-exome sequencing**. Exonic sequences from DNA samples of patients III-3 and V-2 were enriched with the SureSelect Human All Exon 50 Mb Kit (Agilent Technologies). Sequences (100-bp paired-end) were generated on a HiSeq2000 (Illumina). Read alignment and variant calling were performed with DNAnexus using default parameters with the human genome assembly hg19 (GRCh37) as reference.

**Cell culture**. Primary fibroblast cell cultures were established from skin biopsies from two patients and three unrelated controls. Cell cultures were propagated under standard conditions in high glucose Dulbecco's Modified Eagle Medium (DMEM) supplemented with 10% fetal bovine serum (FBS), 2 mM L-glutamine, 100 U/ml penicillin, 100 mg/ml streptomycin, and 1 mM sodium pyruvate (Biological Industries, Israel). All of the experiments were performed with control and patient-derived fibroblasts from identical or highly similar passage numbers and in all cases the passage number did not exceed 15.

**Antibodies**. Commercially obtained antibodies included anti-LAP1 (LS-C288839; LifeSpan Biosciences; dilution 1/2000 WB, 1/100 IF), anti-emerin (PA5-29731; Thermo Fisher Scientific; dilution 1/2000 WB, 1/400 IF), anti-lamin A (ab26300; Abcam; dilution 1/1000), anti-PDI (ab2792; Abcam; dilution 1/100), anti-LULL1 (HPA051849; Sigma-Aldrich; dilution 1/200), anti-tubulin (sc-8035, Santa Cruz Biotechnology; dilution 1/5000), anti-lamin A/C (ab133256, Abcam; dilution 1/10000 WB, 1/500 IF), anti-nuclear pore complex proteins mAb414 (MMS-120P; Covance; dilution 1/300), anti-cleaved caspase 3 (Asp175, #9661; Cell Signaling; dilution 1/400), anti-53BP1 (NB100-304; Novus Biologicals; dilution 1/1000), and Alexa Fluor® 488 anti-H2A.X Phospho (Ser139, #613405; BioLegend; dilution 1/200). Secondary antibodies for indirect immunefluorescence staining were TRITC goat anti-rabbit (111-025-003; Jackson ImmunoResearch; dilution 1/300) and Alexa Fluor 488-donkey anti–mouse IgG (715-545-150; Jackson ImmunoResearch; dilution 1/300). Affinity purified anti-torsinA polyclonal antibody was a kind gift from Christian Schlieker (Yale University, New Haven, CT) and was used at a dilution of 1/2000.

**Cell lysates and western immunoblotting**. To prepare total cell lysates, fibroblasts were grown in 10 cm tissue culture dishes to ~70% confluence. Prior to lysis, cells were washed three times in ice-cold phosphate-buffered saline (PBS) and lysed in 1 ml of RIPA-SDS lysis buffer supplemented with a protease inhibitor cocktail (Pierce 88665; Thermo Fisher Scientific), on ice. Lysates were passed three times through an 18-gauge needle and centrifuged for 5 min at $20{,}000 \times g$ and 4 °C, to precipitate debris. Supernatants were mixed with sodium dodecyl sulfate-polyacrylamide gel electrophoresis (SDS-PAGE) loading buffer. SDS-PAGE and immunoblotting were performed using polyvinylidene difluoride membranes (Millipore) and standard techniques. The quantification of immunoblot band intensities from independently prepared cell lysates was based on normalization to alpha-tubulin and performed with the Gels submenu of ImageJ.

**qPCR analysis**. RNA from fibroblasts was extracted using the RNeasy Mini Kit (Qiagen). RNA concentration was measured and cDNA libraries were constructed using the PrimeScript RT master mix (TAKARA). The synthesized cDNA was amplified using the Fast SYBR green master mix (Applied Biosystems) and results were normalized to the housekeeping gene beta tubulin. Data were analyzed by the StepOne software (Applied Biosystems) and represents the average of three independent biological experiments, each performed with triplicate wells for every sample. Primer pairs used for cDNA amplification were as follows: LAP1 first pair, forward (5′-CCAGGATGCAAAATGACAGCA-3′) and reverse (5′-CTTGTCGGC TGGAGGTT GAT-3′) and second pair forward (5′-ACGTCAACAACTATGGC GGG-3′) and reverse (5′-GGTGACGTACACACCCCATC-3′); emerin forward (5′-TTCCCAGAT GCTGACGCTTT-3′) and reverse (5′-GCGTTCCCTATCCT TGCACT-3′); beta tubulin forward (5′-TGAAGCCACAGGTGGCAAAT-3′) and reverse (5′-AGAGTC CATGGTCCCAGGTT-3′).

**Cellular proliferation and other assays**. Cellular proliferation was tested in a XTT colorimetric assay (Biological Industries). Control and patient-derived primary fibroblasts were plated at 50,000 cells/well in 96-well microplates and followed every 24 h for a total of 5 days. Activated XTT solution was prepared according to the

manufacturer's instructions, added to the cells, and further incubated at 37 °C for 4 h, followed by absorbance measurements at 450 and 650 nm using an enzyme-linked immunosorbent assay plate reader. The reference wavelength absorbance (650 nm) was subtracted to obtain a direct correlation to cellular viability. The apoptosis/necrosis assay kit (ab176749; Abcam) was used to simultaneously monitor apoptotic, necrotic, and healthy cells. For positive cell death controls, the cultures were pre-treated with 10 μM staurosporine or 2 mM $H_2O_2$ for 3 h, switched back to normal medium for 22 h, and further processed for widefield fluorescence microscopy. Cellular senescence was assayed by the senescence β-galactosidase staining kit (#9860; Cell Signaling). Control and patient-derived primary fibroblasts were grown for 5 days in 24-well plates, before fixation and processing for β-galactosidase staining at pH 6, according to the manufacturer's instructions. For the positive starvation-induced senescence control, fibroblasts were grown for 4 days in normal medium and switched to serum-free medium for the last day. Samples for anti-cleaved caspase 3, anti-γH2AX, and anti-53BP1 staining were prepared for direct or indirect IF as detailed below. Pretreatment with 30 μM etoposide for 30 min served as a positive control for the DNA damage response.

**IF and light microscopy**. For indirect IF, cells were grown on poly-lysine-coated coverslips for 24 h, fixed with 3.7% paraformaldehyde for 30 min, permeabilized with PBS/0.1% Triton X-100 for 3 min on ice, and blocked 30 min with PBS/5% FBS, followed by the application of primary and secondary antibodies according to the manufacturer's instructions. Coverslips were mounted in Fluoromount-G (SouthernBiotech). Images were acquired on an Olympus BX61TRF motorized microscope equipped with a DP74 digital camera, using a PlanApo ×100, NA 1.4 oil-immersion objective, or a Zeiss upright Axio Imager.M2 with ApoTome 2.0 equipped with an Orca Flash 4.0 V3 digital CMOS camera (Hamamatsu), using a PlanApochromat ×100, NA 1.4, oil-immersion objective (Carl Zeiss). Confocal and stimulated emission depletion (STED) microscopy were performed on a Leica DMi8 inverted microscope equipped with White Laser technology, 3 depletion lasers (595, 660, and 775 nm), and HyVolution super-resolution imaging, using an HC PlanApo, CS2, ×63, NA 1.4, oil-immersion objective. To quantify the anti-lamin A/C staining intensity, a custom written ImageJ macro was used to outline the nuclear rim in confocal images. This macro was originally developed by Ved P. Sharma (Albert Einstein College of Medicine, New York, version 3, Nov. 30, 2011) to outline the leading edge segment in migrating cells. A segment line was drawn along the edge of anti-PDI staining and a 1-μm wide ring was defined around the perimeter of each nucleus to measure the integrated fluorescence intensity for anti-lamin A/C staining in 30 randomly chosen nuclei for each cell line. Quantitative analysis of the occurrence of cytoplasmic channels in fibroblast nuclei relied on the parallel identification of channels by Hoechst 33258 staining and a NE marker (anti-lamin or mAb414). For the quantification of nuclear morphology parameters, nuclear height was determined in the Leica Application Suite X (defined as the distance between the upper and lower tangential planes in each z-stack). The polygon selection tool in ImageJ was used to delineate the NE and derive measurements for the nuclear perimeter ($P$) and area ($A$). The estimated nuclear volume was calculated as $A \times P$ and nuclear circularity was calculated as previously described[48,49] by the following equation: $C = 4\pi A P^{-2}$.

**Electron microscopy**. For transmission electron microscopy, cells were grown in 10-cm tissue culture dishes to ~70% confluence, and samples were prepared in two different ways to obtain different orientations of the flat cell nuclei. All the fixation and preparation procedures were carried out at room temperature. The cells were washed twice in culture medium without serum and once in the primary fixation solution containing: 2% paraformaldehyde, 2% glutaraldehyde (Electron Microscopy Sciences), 80 mM PIPES KOH, pH 6.8, 1 mM MgCl, 150 mM sucrose. Fresh fixation solution was added for 1 h, followed by three washes in 0.1 M sodium cacodylate, pH 7.4. Cells were scraped off the dish using a rubber cell scrapper and centrifuged in Eppendorf tubes for 5 min at $20{,}000 \times g$. The supernatant was removed and the cell pellet was captured and solidified in low melting point agarose (Thermo Fisher Scientific). Agarose lumps containing the cells were washed three times in cacodylate buffer and post-fixed for 1 h in 1% $OsO_4$ in cacodylate buffer supplemented with 5 mM $CaCl_2$, 0.5% $K_2Cr_2O_7$, and 0.5% $K_4[Fe(CN)_6]$. The samples were then washed in ultra-pure water, stained with 2% uranyl acetate for 1 h, and dehydrated through a graded ethanol series and finally embedded in EMbed 812 resin (Electron Microscopy Sciences) according to the manufacturer's instructions. Alternatively, to obtain top views of the flat nuclei, cells were cultured on 12-mm coverslips, fixed, and post-fixed in the same solutions described above while still attached to the coverslip surface. The embedding resin was solidified directly on the surface of the coverslips and the polymerized blocks were rapidly immersed in liquid nitrogen to detach the coverslips. A monolayer of flat fibroblasts was thus captured on a single plain inside the polymerized resin block. Ultra-thin sections were prepared on a PT-XL ultramicrotome (RMC Boeckeler) and mounted on copper grids. The sections were counterstained with lead citrate and viewed on a Zeiss *Merlin* scanning electron microscope operating in scanning-transmission (STEM) mode with a 4-channel annular STEM detector at 30 kV electron beam energy.

For scanning electron microscopy, cells were grown to 80% confluency, detached by trypsinization, and subjected to hypotonic treatment to expose nuclei as previously described[50]. The samples were gently centrifuged onto the surface of $5 \times 5$ mm$^2$ silicon chips, fixed in the primary fixation solution and 1% $OsO_4$ described above, and

dehydrated through a graded series of ethanol solutions. The samples were then subjected to critical-point drying using a K850 apparatus (Quorum Technologies) and coated with an ~1-nm-thick layer of iridium using a Q150T turbo-pumped sputter coater (Quorum Technologies) and imaged on a Merlin scanning electron microscope (Zeiss) equipped with a secondary electron in-lens detector.

**Cellular motility.** For the wound healing scratch assay, primary skin fibroblast cell lines from one control and two patients were trypsinized, re-suspended in growth medium, and seeded at density of 20,000 cells per well, in 6 replicates for each cell line, in 96-well ImageLock plates (Essen BioScience). Cells were grown overnight forming a monolayer. Uniform 10-μm-wide scratch wounds were created in all the wells with WoundMaker™ and plates were placed in the IncuCyte ZOOM™ apparatus (Essen BioScience). All experiments were performed in six replicates for each category. Images of collective cell spreading were acquired by high-definition phase contrast and red/green fluorescence (for stably transfected cell lines) and recorded every 2 h for a total duration of 126 h using a ×20, NA 0.60 air objective.

For random 2D cell motility analysis, fibroblasts were plated at 20,000 cells/well in 6-well plates and allowed to adhere for 12 h. Images were acquired by phase contrast and red/green fluorescence and recorded every 30 min for a total of 20 h using the IncuCyte™ acquisition software. Trajectory plots, accumulated distance, velocity, and Euclidean distance were calculated using the Chemotaxis and Migration Tool (ibidi GmbH).

**Lentiviral transduction.** Expression constructs and lentiviral particles pseudotyped with VSV-G protein were prepared by GeneCopoeia. Human LAP1B (NM-0011267578.1) and LAP1C (AK021613.1) coding sequences were cloned into the pEZ-Lv225 and pEZ-Lv224, respectively, and validated by full-length sequencing. These multicistronic vectors, driven by the EF1a promoter, were chosen in order to avoid overexpression. The ORF-cloning site is followed by EFGP (in pEZ-Lv225) or mCherry (pEZ-Lv224) and a puromycin resistance gene, interspersed by internal ribosome entry sites. Lentiviral particles were produced and purified by GeneCopoeia using standardized protocols with packaging in 293Ta cells and viral titers determined in the range of $1.7–2.6 \times 10^8$ TU/ml. Control and patient-derived primary fibroblasts were plated at 250,000 cells/well in 6-well plates, grown for 24 h, and transduced with equal amounts of viral particles in 0.5 ml DMEM supplemented with 8 μg/ml poly-brene for 6 h. The cells were then incubated for 3 days in DMEM supplemented with FBS and antibiotic resistance selection was applied by the addition of 1 μg/ml pur-omycin (Sigma Aldrich). After several rounds of antibiotic selection interspersed with growth in regular medium, the successful introduction of each construct was verified in 100% of the cells, as visualized by GFP or mCherry fluorescence in the IncuCyte ZOOM™ apparatus (Essen BioScience) using a ×20, NA 0.60 air objective. In the attempted combined rescue with both LAP1B and LAP1C coding constructs, only low efficiency of stable transfection could be achieved, but GFP and mCherry were both observed in all the cells throughout the trajectory plot analysis.

**Statistical analysis.** Except where otherwise indicated, data are presented as mean ± SEM, and the $P$ values were determined by two-tailed Student's $t$ tests. For the quantitative analysis of channel occurrence in Fig. 3b, a two-proportion $z$-test was used. $P$ values < 0.05 were considered to be statistically significant.

## Data availability

The data that support the findings of this study are available within the paper and its supplementary information files or available from the corresponding authors upon reasonable request. Genetic data have been deposited in the Leiden Open Variation Database, version 3 (http://www.lovd.nl/3.0/home) at the *TOR1AIP1* gene locus under accession number 0000440045.

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

## Acknowledgements

We thank the patients and their families for their participation in this study. We also thank Helena Sabanay for advice on electron microscopy, Basem Hijazi for advice on statistical analysis, Ved P. Sharma for custom written software, Nikola Lukic and Alessandro Genna for advice on single-cell tracking, as well as Michael Blank and Golan Nadav for helpful discussions. This work was supported by a research grant from the Israel Science Foundation (958/15) to A.H.

## Author contributions

B.F., F.Z. and N.B. designed and conducted all the experiments with patient-derived fibroblasts. Y.B. designed and analyzed qPCR experiments. E.C., Z.B.N., O.E. and R.S. analyzed and interpreted the clinical data. A.S. and O.E. analyzed and interpreted genomic data. M.A. performed confocal and STED microscopy. O.E., A.H. and R.S conceived the project and designed experiments. B.F. prepared the figures. A.H. and R.S wrote the manuscript and all the authors commented on it.

## Additional information

**Competing interests:** The authors declare no competing interests.

