## [Peer Review File · Nature Communications]

Reviewer #1 (Remarks to the Author):

Fichtman et al. report a novel nonsense mutation in TOR1AIP1 gene identified in 7 patients with multisystemic symptoms. This nonsense mutation abolishes the expression of all isoforms of the protein in patient-derived fibroblasts, which show defects in the architecture of nuclear envelopes and cell motility.

The author's findings are novel and significant, which will improve our knowledge in the field of nuclear envelopopathies. While a couple of case studies on TOR1AIP1 gene with one or two patients were published previously, the current manuscript describes a novel nonsense mutation in 7 patients from 5 different families. Moreover, unlike previous mutations that generate truncated proteins or defects in only one isoform, this nonsense mutation leads to complete loss of LAP1B and LAP1C expression and causes multiple clinical symptoms indicating essential roles of LAP1 protein encoded by TOR1AIP1 in human development and growth. The author's conclusions are well aligned with their experimental results. In particular, the genetic data for identification of the mutation is convincing and biochemical studies to show the nonsense mutation lead to complete absence of LAP1 are well performed.

I have only a few relatively minor comments that the authors should address:

1. The authors show that a small portion of fibroblasts from patients have cytosolic channels inside their nuclei. It would be more informative to demonstrate whether those cells with defective nuclear architecture are not just undergoing cell death.
2. In the cell motility assay, it would be more convincing if the authors show that the cell proliferation rate has not changed in patient cells.
3. The description of statistical methods is missing in Figure 2, 3 and 6 legends.
4. In the Abstract, the authors should avoid claims of novelty or being first. There is no reason for the authors to in "for the first time in human" in the sentence "Our study describes for the first time in humans the complete absence of both LAP1 isoforms..."
5. On page 5: "Remarkably, over the years patients gradually develop progeroid-like appearance." There is no reason to claim this is remarkable.

Reviewer #2 (Remarks to the Author):

In their manuscript entitled “Combined loss of LAP1B and LAP1C results in a novel early onset multisystemic nuclear envelopathy”, Fichtman et al. describe the identification of a novel non-sense TOR1AIP1 mutation in patients presenting at birth with bilateral cataract, growth retardation, severe progressive neurological impairment, and early lethality. TOR1AIP1 encodes the inner nuclear membrane protein lamina-associated polypeptide 1 (LAP1), and the authors show that fibroblasts isolated from individuals homozygous for the aforementioned mutation lack the expression of the LAP1 isoforms LAP1B and LAP1C. In addition, the authors reveal that patient-derived fibroblasts exhibit nuclear morphology defects (i.e. large nucleus-spanning channels containing trapped cytoplasmic organelles) and inefficient directional migration. Given that TOR1AIP1 mutations were recently demonstrated to result in muscular dystrophy and progressive dystonia with cerebellar atrophy, the results presented in this manuscript further highlight the importance of the loss of LAP1 function during the pathogenesis of several human diseases across a broad clinical spectrum. Despite the significance of describing for the first time the absence of both LAP1B and LAP1C in humans and the resulting clinical phenotypes, there are several major and minor issues that prevent this reviewer from recommending the publication of this manuscript in its current form. These issues are outlined below.

Major Issues:

- 1) LAP1 was first described in Senior and Gerace (1988) J Cell Biol. In this paper, the authors demonstrate the existence of three LAP1 isoforms (LAP1A, LAP1B, and LAP1C) in rat liver nuclear envelopes. Later, Shin et al. (2013) Dev Cell showed that these three LAP1 isoforms were detectable by Western blot with an anti-LAP1 antibody in protein extracts of skeletal muscle and liver from normal adult humans and 6-week old wild type C57/B6 mice. Yet, the manuscript by Fichtman et al. states that there are only two LAP1 isoforms: LAP1B and LAP1C. The authors need to address why they neglected to mention anything about the LAP1A isoform (aside from one sentence in the Discussion).
- 2) The method used to quantify the “lamin nuclear rim intensity” in Figure 3A is poorly described. How exactly was the “intra nuclear” fluorescence intensity determined? In addition, epifluorescence is really not the optimal imaging modality to use for such an analysis. Instead, I would recommend that the authors use a confocal microscope so that they can better identify the nuclear rim and that they co-stain the fibroblasts for an ER marker (i.e. sec-61 β) so that they can identify the nuclear envelope in a manner that is independent of laminA/C staining.
- 3) The authors need to provide some quantification of the nuclear morphology defects that they report in the patient fibroblasts. Specifically, they should quantify nuclear eccentricity, nuclear area, nuclear volume, nuclear height, and the size/area of the nuclear-spanning channels. Otherwise, the results seem rather descriptive.

4) It is completely unclear to me why the authors found it necessary to use STED to image NPCs in Figure 4. Did they also perform NPC STED in control fibroblasts? If so, did the lack of LAP1B/LAP1C affect the average NPC diameter and/or distribution relative to controls?

5) In Figure 5, the authors only show EM images of patient-derived fibroblasts. Why? Did the lack of LAP1B/LAP1C impact the structure of the nuclear envelope (i.e. width of the nuclear envelope, etc.) in patient cells relative to controls?

6) The authors provide data on the ability of patient-derived fibroblasts to close a wound relative to controls in terms of the change in “wound confluence” over time. What is the speed of individual cells migrating into the wounds? Also, do sparsely plated patient-derived fibroblasts display a similar migration defect outside of the context of the wound? Moreover, the authors would do good to mention that Nery et al. (2008) J Cell Sci previously demonstrated that torsinA, which is thought to be activated by LAP1, is required for the efficient directional migration of fibroblasts.

7) The authors should attempt to rescue the nuclear morphology and cell migration defects observed in the patient-derived fibroblasts by re-expressing either LAP1B or LAP1C alone or together. This would solidify whether or not the reported phenotypes were specifically caused by the absence of LAP1 and if so, which particular isoform.

8) Based on their results, the authors suggest that LAP1 “may be affecting the complex network of interactions extending from the nuclear lamina through the NE and to the cytoskeleton”. In addition, they state “LAP1 is a classical candidate for being a NE “toolbox protein” required for nuclear positioning and cell motility”. However, they are definitely not the first to propose such a model (see Atai et al. (2012) Int J Cell Biol as well as Saunders and Luxton (2016) Cell Mol Bioeng). In addition, the authors completely neglect to mention that LAP1 and torsinA were recently demonstrated by Saunders et al. (2017) J Cell Biol to be required for LINC complex-mediated actin-dependent rearward nuclear movement during centrosome orientation in migrating fibroblasts.

Minor Issues:

1) The authors should change “components” to “proteins” at the end of the 1st sentence in the Abstract.

2) Also in the Abstract, the authors should remove the word “and” in between “cytoplasmic organelles” and “traversing the nucleus”.

3) The authors should change “torsin A” to “torsinA” throughout the manuscript.

4) In the Introduction the authors state “LAP1 interacts with several proteins including nuclear lamins, emerin, torsinA and LULL1”; however, to my knowledge there is no evidence in the literature to show that LAP1 interacts with LULL1.

- 5) In the “Electron microscopy reveals large cytoplasmic channels in patient nuclei” section of the Results, the authors need to change “views” to “view” in the sentence that begins “These top views produced typical...”.
- 6) In the Discussion, the authors state “interactions of LAP1 span the perinuclear space and extend to both sides of the NE”. To which interactions are the authors referring exactly?
- 7) Also in the Discussion, the authors need to provide references for the statement made at the end of the 6th paragraph ending in “which depends on cell migration and the colonization of specific developmental niches”.
- 8) Again in the Discussion, the authors should change “is” to “are” in the sentence that starts with “We assume that the lack of myopathy...”.
- 9) In the figure legend for Figure 6, the authors should remove “cell spreading”.
- 10) In the “Immunofluorescence Microscopy” section of the Materials and Methods, the authors need to provide information (i.e. PlanApo, oil-immersion, NA) about the objectives used to generate the images presented in this manuscript.

Reviewer #3 (Remarks to the Author):

The authors describe the homozygous o.Arg321* mutation in TOR1AIP1 in five families with children affected by a severe cerebellar and cortical atrophy leading to microcephaly, cataracts, hearing loss, cachexia, and early demise. The mutations leads to a complete loss of LAP1B and –C, which gives rise to peculiar holes in the nucleus that contain mitochondria and vesicles. Migration of mutated cells is delayed in a scratch wound assay.

General: The manuscript is clearly written and the genetic findings are highly relevant as they demonstrate the severe and of envelopathies related to LAP1 proteins and present a highly unusual nuclear abnormality. However, the mutation description has to be improved and the investigations into the disease mechanism have shortcomings.

Specific:

1. R321X is not an appropriate mutation description. Please use the correct nomenclature p.Arg321*.
2. Figure 2b: Please describe the TOR1AIP1 mutations on the protein level according to nomenclature rules. It is irrelevant whether the already known mutations have been described in Turkish or Moroccan patients. Furthermore, the mutation “M” is at position 482, not 483, and the known mutations p.Pro43Leufs*15 and p.Leu394Pro are not mentioned at all.

3. Figure 2c: Only lamin A is shown, but lamin C is also expressed in fibroblasts and should be included as well.
4. It is surprising that such a profound disruption of the lamina does not result in herniation. The authors should comment on this and state that they have excluded this effect.
5. The authors extensively discuss the important role of the nuclear position for cell migration. Accordingly, a reduction in migration capacity is shown. However, the dramatic phenotype indicates that the cellular defect goes way beyond a reduced cellular migration. It is inadequate that the other well-known effects of alterations of the nuclear lamina are not addressed at all. This is prerequisite for publication in this impact factor range. Cell proliferation, apoptosis, DNA damage, and cellular senescence have to be included in the cellular analysis.

Reviewer #4 (Remarks to the Author):

In their manuscript entitled “Combined loss of LAP1B and LAP1C results in a novel early onset multisystemic nuclear envelopathy,” Fichtman et al. identify a homozygous nonsense mutation in the TOR1AIP1 gene in 7 patients with multiple severe symptoms and early lethality. The authors persuasively show that the disease state likely results from a loss of both LAP1 isoforms and document several nuclear envelope abnormalities that are evident in patient fibroblasts. Importantly, this study represents the first demonstration of a disease resulting from the loss of both major LAP1 isoforms, and it extends the family of diseases associated with mutations in both nuclear envelope resident proteins and Torsin ATPases/Torsin cofactors. Overall, the data are of good quality and the findings are novel (in particular the “nuclear/cytoplasmic channel” phenotype resulting from LAP1 perturbation) and of interest to a broad audience. Therefore, we recommend publication in Nature Communications if the authors can address the following concerns.

Major Concerns:

1. In figure 2C, there appear to be several LAP1 immunoreactive bands in patient samples, including a prominent one of ~35 kDa, that appear only in the “Patient 1” lane. However, on p. 7 of the text, the authors state that they “see no clear evidence for truncated forms of the protein in these patient cell lysates.” Could the authors address why they don’t acknowledge these smaller bands in the text? A longer exposure may also be helpful (could be included in the supplements) for readers to decipher whether there are any other LAP1 truncation products present in patient cells. In the opinion of this reviewer, it is quite possible that protein turnover also contributes to the reduction of LAP1 levels (cf. Tsai et al. PMID: 27336722) since the observed LAP1 reduction on the protein level (reduced by at least 20-fold, Fig. 2C) is more pronounced than the reduction on the RNA level (reduced to ~40%, Fig. 2D, Suppl. Fig. 2b). This possibility should be acknowledged in the text (e.g. along with RNA decay in 1st paragraph of discussion), but does not necessarily have to be addressed by experiment (though stabilization of LAP1 mutant derivatives upon addition of proteasome inhibitors to patient fibroblasts could shed light on this point and would be easy to do).

2. In figure 3, the authors argue that Lamin A/C staining is less intense around the nuclear rim than in control cells. However, their decision to use a widefield microscope instead of a confocal for this analysis makes this firm conclusion somewhat difficult to justify. In this reviewer's experience, changing the focus in widefield microscopy could cause a protein to appear more or less around the nuclear periphery. Also, if there is an available marker (such as MAb414 or Emerin) that does not change to a more intranuclear location, this could be used as a counterstain. As it stands, it is not convincing that Lamin A/C is truly more intranuclear than around the nuclear rim, but using a confocal microscope and an appropriate counterstain should resolve this issue.

3. Given that the channel phenotype demonstrated in figure 5 is very interesting and novel, the authors should give the reader a better idea of how penetrant this phenotype is. We recommend performing a statistical analysis of control versus patient fibroblast to demonstrate how often these structures are seen in EM cross-sections (or, if easier, use immunofluorescence with $n = 100$ each and statistical analysis). It would also be useful to mark the channel in IF images by an arrowhead since, at least for the Hoechst stain, channels could be confused with nucleoli. Perhaps the authors could point out that the other stains (Lamin, Mb414) are a better diagnostic tool since in those cases channels cannot be confused with nucleoli.

Minor Concerns:

1. On page 6, the sentence starting with "33 and 8 homozygous variants remained..." is unclear and should be rephrased. Nonetheless, from this reviewer's understanding, there were 8 homozygous variants in these patients that were not predicted to be benign. The authors then state that "only one variant c.961C>T was predicted to be pathogenic." It is not clear how pathogenicity was predicted in this analysis, so it is hard to assess whether other genes may still be candidates for the phenotypes presented here. Perhaps including a supplementary table with these other remaining variants and/or explaining more detail about why they were eliminated would be more convincing for the reader.

2. Several times in the paper (pages 3 and 7, figure legend for supplementary figure 2), the authors describe LULL1 as a LAP1 interaction partner. To this reviewer's knowledge, no direct interaction between LAP1 and LULL1 has been reported, and the referenced papers do not provide evidence of an interaction. Instead, LAP1 and LULL1 have highly similar luminal domains and both independently bind and activate the ATPase activity of TorsinA or TorsinB (Zhao et al, PMID 23569223). This should be corrected accordingly.

3. In figure 4, a control fibroblast should be imaged and shown beside the patient fibroblast.

4. Interestingly, TorsinA knockout MEFs and Lap1-silenced fibroblasts were previously also shown to have delayed migration in a wound-healing assay (Nery et al, PMID 18827015; Saunders et al., PMID:28242745). Given the relationship between TorA and LAP1, the authors could cite these studies in the text.

5. On page 13 of the discussion, the authors claim that “the exceptionally severe early onset phenotype experienced in our patients is attributed to a deficiency of LAP1C rather than LAP1B.” However, there is no known case of LAP1C deletion without also having LAP1B deletion. Therefore, it is not possible to determine whether the phenotypes presented here are the result of an additive effect of losing both LAP1 isoforms or if the phenotypes are truly unique to a loss of LAP1C. Without having access to patients or cells lacking only LAP1C, the authors should downscale this conclusion.

6. On several occasions the authors refer to a “dramatic” reduction of LAP1 on the RNA level. It’d be better to avoid the term “dramatic” and instead state a numerical value (e.g., “was reduced by X %” or “by x-fold”)

7. Interestingly, it appears that TorA is processed (cf. Zhao et al. PMID:26953341) specifically in patient-derived samples (suppl. Fig. 2A). This does not need to be experimentally addressed but it might be worth mentioning in the text.

8. The observation that Emerin is upregulated in patients is quite interesting and fits well to the observation of synthetic lethality between LAP1 and emerin in mouse models (Shin et al. 24055652). Perhaps this could be mentioned in the text more explicitly (the citation was noted).

NCOMMS-18-09063 Revision - Point-by-point response

General summary of changes in the figures

Article figures:

Figure 2b - an amended version of the scheme describing LAP1 isoforms and mutations.

Figure 3 has been changed and includes a new **panel a** showing anti-lamin nuclear rim intensity measurements from confocal images. Panel b has been re-arranged in space and includes a new bar-chart (bottom-right) with the quantitative analysis of channel occurrence.

Figure 4 has an additional column (left) showing a control nucleus, as well as zoomed-in versions of the side panels in the z-stack column (right). The former STED mode inset has been moved to new Supp. Figure 6c.

Former **Figure 6** is now (unchanged) Figure 6a. New **panel b** shows the additional analysis of single cell trajectory plots and motility parameters outside the context of a wound.

New **Figure 7** including 3 panels describes rescue experiments by transduction of lentiviral vectors expressing LAP1B and LAP1C.

Supplementary figures:

Supp. Figure 2 includes new **panels c & d** showing an additional immunoblot for lamin A/C and a longer exposure for the original blot from the top of Figure 2c.

Previous Supp. Figures 3 and 4 have been merged into the new Supp. Figure 3.

Supp. Figures 4-10 are **all new**.

Supp. Figure 4 - shows EM images of control fibroblasts, as well as an additional example of a patient-derived fibroblast nucleus containing minute suspected channels.

Supp. Figure 5 - shows a comparison of nuclear morphology parameters measured by light microscopy in control and patient-derived fibroblasts.

Supp. Figure 6 - shows the distribution of NPCs in control & patient-derived fibroblasts as determined by confocal microscopy and scanning EM.

Supp. Figure 7 - compares the proliferation rate of control and patient-derived fibroblasts.

Supp. Figure 8 - shows staining with fluorescent markers for apoptosis, necrosis and cellular senescence.

Supp. Figure 9 - shows staining for the DNA damage response markers γ H2AX and 53BP1.

Supp. Figure 10 - shows additional information for the lentiviral rescue experiments described in new Figure 7: immunoblot analysis for the expression levels of the LAP1 isoforms and an attempt for combined rescue with both vectors.

Response to Reviewer 1

Reviewer #1 (Remarks to the Author):

Fichtman et al. report a novel nonsense mutation in TOR1AIP1 gene identified in 7 patients with multisystemic symptoms. This nonsense mutation abolishes the expression of all isoforms of the protein in patient-derived fibroblasts, which show defects in the architecture of nuclear envelopes and cell motility.

The author's findings are novel and significant, which will improve our knowledge in the field of nuclear envelopopathies. While a couple of case studies on TOR1AIP1 gene with one or two patients were published previously, the current manuscript describes a novel nonsense mutation in 7 patients from 5 different families. Moreover, unlike previous mutations that generate truncated proteins or defects in only one isoform, this nonsense mutation leads to complete loss of LAP1B and LAP1C expression and causes multiple clinical symptoms indicating essential roles of LAP1 protein encoded by TOR1AIP1 in human development and growth. The author's conclusions are well aligned with their experimental results. In particular, the genetic data for identification of the mutation is convincing and biochemical studies to show the nonsense mutation lead to complete absence of LAP1 are well performed.

I have only a few relatively minor comments that the authors should address:

1. The authors show that a small portion of fibroblasts from patients have cytosolic channels inside their nuclei. It would be more informative to demonstrate whether those cells with defective nuclear architecture are not just undergoing cell death.

We have tested this possibility by adding specific assays testing apoptosis, necrosis and senescence (see also: Reviewer 3, comment#5 regarding the known effects of other severe laminopathies). The text in the middle of page 13 relates to this and the results are shown in Supp. Figure 8. We note that we found no correlation between the few patient-derived cells staining with cellular death markers under normal conditions and the presence of cytoplasmic channels.

2. In the cell motility assay, it would be more convincing if the authors show that the cell proliferation rate has not changed in patient cells.

Agreed. This is now shown in Supp. Figure 7. If anything, the proliferation rate is slightly higher in patient cells compared to the controls and this cannot explain the defects in cell motility.

3. The description of statistical methods is missing in Figure 2, 3 and 6 legends.

Details of statistical methods have now been added to the legends and when necessary to the relevant Methods sub-sections. Please note that parts of the mentioned figures have now changed (3a has been replaced, bar chart added to 3b, 6 has new panel b)

4. In the Abstract, the authors should avoid claims of novelty or being first. There is no reason for the authors to in “for the first time in human” in the sentence “Our study describes for the first time in humans the complete absence of both LAP1 isoforms...”

The text in the last part of the abstract has been amended accordingly.

5. On page 5: “Remarkably, over the years patients gradually develop progeroid-like appearance.” There is no reason to claim this is remarkable.

The word “Remarkably” has been removed from the text.

Response to Reviewer 2

Reviewer #2 (Remarks to the Author):

In their manuscript entitled “Combined loss of LAP1B and LAP1C results in a novel early onset multisystemic nuclear envelopathy”, Fichtman et al. describe the identification of a novel non-sense TOR1AIP1 mutation in patients presenting at birth with bilateral cataract, growth retardation, severe progressive neurological impairment, and early lethality. TOR1AIP1 encodes the inner nuclear membrane protein lamina-associated polypeptide 1 (LAP1), and the authors show that fibroblasts isolated from individuals homozygous for the aforementioned mutation lack the expression of the LAP1 isoforms LAP1B and LAP1C. In addition, the authors reveal that patient-derived fibroblasts exhibit nuclear morphology defects (i.e. large nucleus-spanning channels

containing trapped cytoplasmic organelles) and inefficient directional migration. Given that TOR1AIP1 mutations were recently demonstrated to result in muscular dystrophy and progressive dystonia with cerebellar atrophy, the results presented in this manuscript further highlight the importance of the loss of LAP1 function during the pathogenesis of several human diseases across a broad clinical spectrum. Despite the significance of describing for the first time the absence of both LAP1B and LAP1C in humans and the resulting clinical phenotypes, there are several major and minor issues that prevent this reviewer from recommending the publication of this manuscript in its current form. These issues are outlined below.

Major Issues:

1) LAP1 was first described in Senior and Gerace (1988) J Cell Biol. In this paper, the authors demonstrate the existence of three LAP1 isoforms (LAP1A, LAP1B, and LAP1C) in rat liver nuclear envelopes. Later, Shin et al. (2013) Dev Cell showed that these three LAP1 isoforms were detectable by Western blot with an anti-LAP1 antibody in protein extracts of skeletal muscle and liver from normal adult humans and 6-week old wild type C57/B6 mice. Yet, the manuscript by Fichtman et al. states that there are only two LAP1 isoforms: LAP1B and LAP1C. The authors need to address why they neglected to mention anything about the LAP1A isoform (aside from one sentence in the Discussion).

We apologize for this omission. The introduction has been changed and now mentions the three rat and mouse isoforms A, B and C (page 3). We later (briefly) mention the open question regarding the potential human equivalent to the higher mw LAP1A (pages 7-8). We note that we originally referred to “at least two functional LAP1 isoforms...in humans...from a single gene” and even the cited references (e.g., 15, 20, 42) give different accounts of the situation. In the end, we can only relate to what is clearly detectable in fibroblasts by the antibodies available to us. We note that in addition to the LS-C288839; LifeSpan antibody presented in the manuscript we tried: LS-C353382; LifeSpan and ab2737 [RL13]; Abcam (no immunoblot detection) and we attempted to obtain previously published antibodies from Larry Gerace, La Jolla USA (copy of e-mail shown below) and one of the corresponding authors of Shin et al., 2013 (ref#18; we received no response).

Given this situation, we think it is fair to say that our study focuses on the two major isoforms that we can clearly identify in human fibroblasts. Never the less, we have slightly modified the text in several places: e.g., at the end of the abstract, in the 3rd sub-heading of the Results on page 7, and at the top of page 19: “a mutation that abolishes the expression of both of the major LAP1 isoforms in humans”.

We also added the following sentence on page 8 at the end of the first immunoblot description: “We did not detect any sign of a higher molecular weight isoform, but note that the unique position of the nonsense mutation, preceding the transmembrane segment, would also be predicted to affect such a product of the *TOR1AIP1* gene”.

2) The method used to quantify the “lamin nuclear rim intensity” in Figure 3A is poorly described. How exactly was the “intra nuclear” fluorescence intensity determined? In addition, epifluorescence is really not the optimal imaging modality to use for such an analysis. Instead, I would recommend that the authors use a confocal microscope so that they can better identify the nuclear rim and that they co-stain the fibroblasts for an ER marker (i.e. sec-61 β) so that they can identify the nuclear envelope in a manner that is independent of laminA/C staining.

See also: Reviewer 4, comment #2.

Epifluorescence has been replaced with confocal microscopy and co-staining for PDI as an ER marker, as suggested - in Figure 3a. The quantification of the signal is explained in the figure legend and in more detail in the methods section. We thank both reviewers for this helpful comment.

3) The authors need to provide some quantification of the nuclear morphology defects that they report in the patient fibroblasts. Specifically, they should quantify nuclear eccentricity, nuclear area, nuclear volume, nuclear height, and the size/area of the nuclear-spanning channels. Otherwise, the results seem rather descriptive.

The nuclear-spanning channels are highly irregular in shape, as observed both by light and electron microscopy. This is now stated in the text in the middle of page 10, together with measurements of apparent dimensions by LM. Again, the text in the next section describes these features as observed by side- and top-views by TEM. Importantly, the new Supp. Figure 4b also shows examples of extremely small channels in cross-section (see Reviewer 4, comment#3).

Other measured parameters of nuclear morphology are now presented under a new sub-heading in the Results section on page 12 and in Supp. Figure 5. We think it is important to describe these features, even though they do not add up to the dramatic changes in morphology reported for some other laminopathies. Specifically, for circularity (the reverse of eccentricity): we found no significant difference between patient and control nuclei and this is in strong contrast to HGPS and other laminopathies. The differences in relation to other laminopathies are also discussed on page 19. See also: Reviewer 3, comments #4-5.

4) It is completely unclear to me why the authors found it necessary to use STED to image NPCs in Figure 4. Did they also perform NPC STED in control fibroblasts? If so, did the lack of LAP1B/LAP1C affect the average NPC diameter and/or distribution relative to controls?

See also: Reviewer 4, minor comment #3.

We agree with both reviewers that the description of this experiment was confusing. mAb414 staining of a control cell nucleus is now shown on the left of Figure 4 and the text has been amended on page 10. The only reason to include the enlarged STED imaging panel was to demonstrate the specificity of the antibody (a trivial fact for researchers in the nuclear transport field, but possibly useful for others). We have now moved this panel to Supp. Figure 6 which concerns NPCs.

Confocal microscopy was used in Figure 4 because of the improved resolution in the z-axis, allowing us to follow a cytoplasmic channel from top-to-bottom through the flat nucleus (a z-stack is only shown for the patient cell because there are no channels in control cells - and there is nothing to follow).

The new Supplementary Figure 6 shows by both confocal and scanning electron microscopy (SEM) that there is no difference between control and patient cells in the distribution of NPCs in the NE. NPC diameter and general features also appeared normal in both samples, as judged by SEM.

5) In Figure 5, the authors only show EM images of patient-derived fibroblasts. Why? Did the lack of LAP1B/LAP1C impact the structure of the nuclear envelope (i.e. width of the nuclear envelope, etc.) in patient cells relative to controls?

This is similar to the previous remark on Figure 4. In this case, we feel that adding EM images of control nuclei to Figure 5 itself would distract from the main issue, which is a detailed high-resolution analysis of the channels in patient cell nuclei. The requested images of control nuclei (including side- and top-views and higher magnifications of the NE) are shown in Supp. Figure 4a and referred to in the text on page 11. Specifically, we see no evidence for herniation or changes in NE width. See also: Reviewer 3, comment #4.

6) The authors provide data on the ability of patient-derived fibroblasts to close a wound relative to controls in terms of the change in “wound confluence” over time. What is the speed of individual cells migrating into the wounds? Also, do sparsely plated patient-derived fibroblasts display a similar migration defect outside of the context of the wound? Moreover, the authors would do good to mention that Nery et al. (2008) *J Cell Sci* previously demonstrated that torsinA, which is thought to be activated by LAP1, is required for the efficient directional migration of fibroblasts.

The wound closing assay follows the collective movement of a dense population of cells. Formally, the slope of the curve shown in Figure 6a represents the collective speed of the cells, but this is not a very useful measure, since cells collide and even climb over each other, as can be seen in the Supplementary movies.

We followed the suggestion of the reviewer and tested the migration properties of individual, sparsely plated fibroblasts and the results - now shown in Figure 6b - demonstrate a strong functional effect of the mutation.

The reference - *Nery et al. (2008) J Cell Sci* - has been added and is now mentioned on pages 4, 14 and 18. See also: Reviewer 4, minor comment #4.

7) The authors should attempt to rescue the nuclear morphology and cell migration defects observed in the patient-derived fibroblasts by re-expressing either LAP1B or LAP1C alone or together. This would solidify whether or not the reported phenotypes were specifically caused by the absence of LAP1 and if so, which particular isoform.

We thank the reviewer for this important suggestion. We chose to use stable transfection by multicistronic lentiviral vectors and the EF1a promoter (to avoid potential artifacts of overexpression). The results are shown in the new Figure 7 and described in a new section at the end of the Results on pages 15-16. We demonstrate substantial rescue of 3 distinct cellular phenotypes: the occurrence of cytoplasmic channels, collective cell migration and trajectory plots (including motility parameters) of individual cells. Although we observed some differences in the extent of rescue between the LAP1B and LAP1C coding constructs, especially regarding the Euclidean distance in single cell motility, we cannot reach a firm conclusion regarding differential roles of the 2 isoforms. This is mostly because of the different level of expression achieved by the 2 constructs (Supp. Figure 10).

8) Based on their results, the authors suggest that LAP1 “may be affecting the complex network of interactions extending from the nuclear lamina through the NE and to the cytoskeleton”. In addition, they state “LAP1 is a classical candidate for being a NE “toolbox protein” required for nuclear positioning and cell motility”. However, they are definitely not the first to propose such a model (see Atai et al. (2012) Int J Cell Biol as well as Saunders and Luxton (2016) Cell Mol Bioeng). In addition, the authors completely neglect to mention that LAP1 and torsinA were recently demonstrated by Saunders et al. (2017) J Cell Biol to be required for LINC complex-mediated actin-dependent rearward nuclear movement during centrosome orientation in migrating fibroblasts.

Agreed. We did not mean to imply we were the first to suggest such a model. Indeed, we cited 2 general references in this context and have now added *Atai et al.*, and *Saunders & Luxton*. We also expanded the discussion on page 18 and mentioned the results of *Saunders et al., 2017*. See also: Reviewer 4, minor comment #4.

Minor Issues:

1) The authors should change “components” to “proteins” at the end of the 1st sentence in the Abstract.

This has now been corrected.

2) Also in the Abstract, the authors should remove the word “and” in between “cytoplasmic organelles” and “traversing the nucleus”.

Corrected.

3) The authors should change “torsin A” to “torsinA” throughout the manuscript.

This has now been changed in the manuscript text and in Supplementary Figure 2.

4) In the Introduction the authors state “LAP1 interacts with several proteins including nuclear lamins, emerin, torsinA and LULL1”; however, to my knowledge there is no evidence in the literature to show that LAP1 interacts with LULL1.

See also: Reviewer 4, minor comment #2.

We agree and apologize for this oversight. The text in the introduction at the end of page 3 now states: “The highly similar luminal domains of LAP1 and LULL1 have both been shown to independently bind and activate torsin ATPases”.

5) In the “Electron microscopy reveals large cytoplasmic channels in patient nuclei” section of the Results, the authors need to change “views” to “view” in the sentence that begins “These top views produced typical...”.

“View” has been changed to “views” and the sentence now makes sense.

6) In the Discussion, the authors state “interactions of LAP1 span the perinuclear space and extend to both sides of the NE”. To which interactions are the authors referring exactly?

We are sorry for this mistake. The text has been amended both at the top of page 4 and in the Discussion at the top of page 18, to reflect the broader sense of dynamic and indirect interactions spanning the NE.

7) Also in the Discussion, the authors need to provide references for the statement made at the end of the 6th paragraph ending in “which depends on cell migration and the colonization of specific developmental niches”.

Three relevant references have been added.

8) Again in the Discussion, the authors should change “is” to “are” in the sentence that starts with “We assume that the lack of myopathy...”.

This sentence has also been changed according to Reviewer 4, minor comment #5, but this error has additionally been corrected.

9) In the figure legend for Figure 6, the authors should remove “cell spreading”.

Corrected; the text now reads: “...showing the initial position (t=0) and the extent of wound closure after 52 hours”.

10) In the “Immunofluorescence Microscopy” section of the Materials and Methods, the authors need to provide information (i.e. PlanApo, oil-immersion, NA) about the objectives used to generate the images presented in this manuscript.

This information has been added on page 25.

Response to Reviewer 3

Reviewer #3 (Remarks to the Author):

The authors describe the homozygous o.Arg321* mutation in TOR1AIP1 in five families with children affected by a severe cerebellar and cortical atrophy leading to microcephaly, cataracts, hearing loss, cachexia, and early demise. The mutations leads to a complete loss of LAP1B and -C, which gives rise to peculiar holes in the nucleus that contain mitochondria and vesicles. Migration of mutated cells is delayed in a scratch wound assay.

General: The manuscript is clearly written and the genetic findings are highly relevant as they demonstrate the severe and of envelopathies related to LAP1 proteins and present a highly unusual nuclear abnormality. However, the mutation description has to

be improved and the investigations into the disease mechanism have shortcomings.
Specific:

1. R321X is not an appropriate mutation description. Please use the correct nomenclature p.Arg321*.

This has been corrected in the text in the middle of page 7 and in Figure 2b.

2. Figure 2b: Please describe the TOR1AIP1 mutations on the protein level according to nomenclature rules. It is irrelevant whether the already known mutations have been described in Turkish or Moroccan patients. Furthermore, the mutation “M” is at position 482, not 483, and the known mutations p.Pro43Leufs*15 and p.Leu394Pro are not mentioned at all.

All these changes have been introduced to Figure 2b and its legend, as well as the Discussion (page 19). We note however, that the “Turkish” and “Moroccan” designations have been used by other authors in previous publications (references 23, 24, 42) and ethnicity is important, especially for physicians who may encounter new suspected cases. Therefore, we have left unchanged the textual reference to Palestinian ancestry (bottom of page 4) and a Palestinian founder mutation (middle of page 7).

3. Figure 2c: Only lamin A is shown, but lamin C is also expressed in fibroblasts and should be included as well.

A second antibody, recognizing both lamins A and C, has been added and the Western blot is shown in Supp. Figure 2c.

4. It is surprising that such a profound disruption of the lamina does not result in herniation. The authors should comment on this and state that they have excluded this effect.

See also: Reviewer 2, comment #5.

We agree, but herniation or nuclear blebbing of the type described for several laminopathies is easy to spot by LM or TEM. We see no evidence for such severe phenotypes and in fact, the NE appears normal in TEM images from patient cells (in the nuclear periphery and around the cytoplasmic channels). This is now stated on pages 11, 12 and 19.

This comparison to other laminopathies is also relevant to the next comment.

5. The authors extensively discuss the important role of the nuclear position for cell migration. Accordingly, a reduction in migration capacity is shown. However, the dramatic phenotype indicates that the cellular defect goes way beyond a reduced cellular migration. It is inadequate that the other well-known effects of alterations of the nuclear lamina are not addressed at all. This is prerequisite for publication in this impact factor range. Cell proliferation, apoptosis, DNA damage, and cellular senescence have to be included in the cellular analysis.

We have added specific assays for all of these aspects and a new sub-heading in the Results section: “The single cell phenotypes differ from known hallmarks of other laminopathies” (page 12). Results are shown in Supplementary Figures 7, 8 and 9 and they differ from previous reports, sometimes showing mild alterations in the opposite direction (higher proliferation rate, lower senescence). We believe the DNA damage response is only a mild one, possibly due to an indirect effect on the lamina.

As mentioned above, this also relates to Reviewer 2, comment #3 and the measurements of nuclear morphology parameters. Overall, our patient-derived fibroblasts do not show the type of defects reported for HGPS and other severe laminopathies. This is summarized on page 19.

Response to Reviewer 4

Reviewer #4 (Remarks to the Author):

In their manuscript entitled “Combined loss of LAP1B and LAP1C results in a novel early onset multisystemic nuclear envelopathy,” Fichtman et al. identify a homozygous nonsense mutation in the TOR1AIP1 gene in 7 patients with multiple severe symptoms and early lethality. The authors persuasively show that the disease state likely results from a loss of both LAP1 isoforms and document several nuclear envelope abnormalities that are evident in patient fibroblasts. Importantly, this study represents the first demonstration of a disease resulting from the loss of both major LAP1 isoforms, and it extends the family of diseases associated with mutations in both nuclear envelope resident proteins and Torsin ATPases/Torsin cofactors. Overall, the data are of good quality and the findings are novel (in particular the “nuclear/cytoplasmic channel” phenotype resulting from LAP1 perturbation) and of interest to a broad audience. Therefore, we recommend publication in Nature Communications if the authors can address the following concerns.

Major Concerns:

1. In figure 2C, there appear to be several LAP1 immunoreactive bands in patient samples, including a prominent one of ~35 kDa, that appear only in the “Patient 1” lane. However, on p. 7 of the text, the authors state that they “see no clear evidence for truncated forms of the protein in these patient cell lysates.” Could the authors address why they don’t acknowledge these smaller bands in the text? A longer exposure may also be helpful (could be included in the supplements) for readers to decipher whether there are any other LAP1 truncation products present in patient cells. In the opinion of this reviewer, it is quite possible that protein turnover also contributes to the reduction of LAP1 levels (cf. Tsai et al. PMID: 27336722) since the observed LAP1 reduction on the protein level (reduced by at least 20-fold, Fig. 2C) is more pronounced than the reduction on the RNA level (reduced to ~40%, Fig. 2D, Suppl. Fig. 2b). This possibility should be acknowledged in the text (e.g. along with RNA decay in 1st paragraph of discussion), but does not necessarily have to be addressed by experiment (though stabilization of LAP1 mutant derivatives upon addition of proteasome inhibitors to patient fibroblasts could shed light on this point and would be easy to do).

Agreed. The text at the bottom of page 7 has been changed accordingly, with a call-out to a longer exposure of this blot in Supp. Figure 2d (potential truncated forms, i.e., smaller immunoreactive bands are marked by red asterisks) and again at the top of page 9. The Discussion (page 17) has been amended to acknowledge the possible contribution of protein turnover.

2. In figure 3, the authors argue that Lamin A/C staining is less intense around the nuclear rim than in control cells. However, their decision to use a widefield microscope instead of a confocal for this analysis makes this firm conclusion somewhat difficult to justify. In this reviewer’s experience, changing the focus in widefield microscopy could cause a protein to appear more or less around the nuclear periphery. Also, if there is an available marker (such as MAb414 or Emerin) that does not change to a more intranuclear location, this could be used as a counterstain. As it stands, it is not convincing that Lamin A/C is truly more intranuclear than around the nuclear rim, but using a confocal microscope and an appropriate counterstain should resolve this issue.

Agreed and done. See: Reviewer 2, comment #2.

3. Given that the channel phenotype demonstrated in figure 5 is very interesting and novel, the authors should give the reader a better idea of how penetrant this phenotype

is. We recommend performing a statistical analysis of control versus patient fibroblast to demonstrate how often these structures are seen in EM cross-sections (or, if easier, use immunofluorescence with n = 100 each and statistical analysis). It would also be useful to mark the channel in IF images by an arrowhead since, at least for the Hoechst stain, channels could be confused with nucleoli. Perhaps the authors could point out that the other stains (Lamin, Mb414) are a better diagnostic tool since in those cases channels cannot be confused with nucleoli.

(* These remarks are related to Figures 3 and 5)

The suggested statistical analysis of cytoplasmic channel occurrence has been added to the last section of Figure 3. This analysis or “channel scoring” has been performed by light microscopy, since we don’t have the means of analyzing large arrays of serial thin-sections by TEM. However, we did add one more example of a TEM section with enlarged regions from a patient cell nucleus in Supp. Figure 4b. This includes examples of extremely small channels in cross-section, that would be below the resolution limit for identification by LM. This argues that the occurrence calculated by LM may be an underestimate. The only caveat is that we cannot exclude the possibility that some of these structures represent invaginations of the NE rather than complete channels traversing the nucleus (text on page 12). The channels in the IF images in Figure 3 are now marked by arrowheads, as suggested. We agree with the reviewer Hoechst staining alone is not the best diagnostic tool for identifying channels and this is now mentioned in the text in the middle of page 10.

Minor Concerns:

1. On page 6, the sentence starting with “33 and 8 homozygous variants remained...” is unclear and should be rephrased. Nonetheless, from this reviewer’s understanding, there were 8 homozygous variants in these patients that were not predicted to be benign. The authors then state that “only one variant c.961C>T was predicted to be pathogenic.” It is not clear how pathogenicity was predicted in this analysis, so it is hard to assess whether other genes may still be candidates for the phenotypes presented here. Perhaps including a supplementary table with these other remaining variants and/or explaining more detail about why they were eliminated would be more convincing for the reader.

We agree with the reviewer that this description was confusing, but there is no doubt about the genetic analysis and the elimination of the other candidates. This is because whole exome sequencing (WES) was performed for 2 patients coming from separate, unrelated pedigrees. This is now stated in the opening paragraph describing WES analysis on page 6 and the rest of the corrected text reads:

“After filtering, 33 and 8 homozygous variants remained for patient III-3 and patient V-2, respectively. Only one potentially pathogenic variant, c.961C>T in *TOR1AIP1*, was shared between both patients, which were very similar phenotypically.”

Pathogenicity was predicted using Mutation Taster, as mentioned in the text, but the real value of such predictions can only be tested by biochemistry and cell biology, as subsequently described. Again, we note that the unambiguous genetic identification of the mutation was solidified by the finding that all 7 patients from 5 separate pedigrees were homozygous for it and the perfect co-segregation of the affected state in all the families (top of page 7).

2. Several times in the paper (pages 3 and 7, figure legend for supplementary figure 2), the authors describe LULL1 as a LAP1 interaction partner. To this reviewer’s knowledge, no direct interaction between LAP1 and LULL1 has been reported, and the referenced papers do not provide evidence of an interaction. Instead, LAP1 and LULL1 have highly similar luminal domains and both independently bind and activate the ATPase activity of TorsinA or TorsinB (Zhao et al, PMID 23569223). This should be corrected accordingly.

See also: Reviewer 2, minor comment #4.

Agreed and corrected on pages 3 and 8 and in the Supp. Figure 2 legend. The reference by Zhao et al., 2013 has been added.

3. In figure 4, a control fibroblast should be imaged and shown beside the patient fibroblast.

See also: Reviewer 2, comment #4.

mAb414 staining of a control cell nucleus is now shown on the left side of Figure 4.

4. Interestingly, TorsinA knockout MEFs and Lap1-silenced fibroblasts were previously also shown to have delayed migration in a wound-healing assay (Nery et al, PMID 18827015; Saunders et al., PMID:28242745). Given the relationship between TorA and LAP1, the authors could cite these studies in the text.

See also: Reviewer 2, comments #6 and #8.

Both references are now cited on page 14, as a prelude to the experiments investigating cell motility in patient fibroblasts and are mentioned again in the Discussion (page 18).

5. On page 13 of the discussion, the authors claim that “the exceptionally severe early onset phenotype experienced in our patients is attributed to a deficiency of LAP1C rather than LAP1B.” However, there is no known case of LAP1C deletion without also having LAP1B deletion. Therefore, it is not possible to determine whether the phenotypes presented here are the result of an additive effect of losing both LAP1 isoforms or if the phenotypes are truly unique to a loss of LAP1C. Without having access to patients or cells lacking only LAP1C, the authors should downscale this conclusion.

We accept the criticism and have downscaled the conclusion. This speculative interpretation is based on the clinical features of the known mutations and the published expression profiles of the 2 isoforms. This is now stated more clearly (top of page 20) and the closing paragraph on page 21 has also been modified.

6. On several occasions the authors refer to a “dramatic” reduction of LAP1 on the RNA level. It’d be better to avoid the term “dramatic” and instead state a numerical value (e.g., “was reduced by X %” or “by x-fold”)

The term “dramatic” has been removed on pages 8-9 and page 17; a numerical value has been added: “<40% of the controls” (bottom of page 8).

7. Interestingly, it appears that TorA is processed (cf. Zhao et al. PMID:26953341) specifically in patient-derived samples (suppl. Fig. 2A). This does not need to be experimentally addressed but it might be worth mentioning in the text.

This is now mentioned in the text on page 8 and in the corresponding Supp. Figure 2a legend. The reference by Zhao et al., has been added.

8. The observation that Emerin is upregulated in patients is quite interesting and fits well to the observation of synthetic lethality between LAP1 and emerin in mouse models (Shin et al. 24055652). Perhaps this could be mentioned in the text more explicitly (the citation was noted).

Agreed. This is now specifically mentioned in the Discussion on page 17.

Reviewer #1 (Remarks to the Author):

This is a revised manuscript by Fichtman et al., reporting a novel nonsense mutation in the TOR1AIP1 gene in 7 patients with multisystemic symptoms. The authors addressed all comments requested by this reviewer and improved the initial version of the manuscript substantially. They performed new experiments that have now included as supplemental figures 7 and 8. The authors also amended the text accordingly.

The data presented in the revised version are of good quality and the author's findings are novel. Therefore, I recommend publication in Nature Communications.

Reviewer #2 (Remarks to the Author):

Overall, I feel that the authors have successfully addressed my previously raised concerns. That being said, I do have a few remaining suggestions that I would like to see addressed before this manuscript can be accepted for publication.

1) In response to the authors' response to my 5th Major Concern, they state "adding EM images of control nuclei to Figure 5 itself would distract from the main issue". I strongly disagree with this statement, as including the control images next to the patient fibroblasts is critical for comparing the nuclear envelope defects observed in the absence of LAP1B and LAP1C.

2) The authors should change the title "Transfection of LAP1-coding constructs into patient-derived fibroblasts rescues multiple cellular phenotypes" to "Expression of LAP1-encoding constructs into patient-derived fibroblasts rescues multiple cellular phenotypes". They are using lentiviral transduction, not transfection, to express the LAP1-encoding constructs in the patient-derived fibroblasts.

3) In the Discussion, the authors state that torsinA is "another direct interaction of the LINC complex"; however, this statement is not correct. While torsinA was shown to have affinity for the KASH domains of nesprins 1-3 by Nery et al. (2008) J Cell Sci, this does not mean that torsinA can interact with an assembled LINC complex. To my knowledge, no one has ever shown that torsinA can directly interact with a LINC complex.

4) The authors should indicate the statistical significance, or lack thereof, in the graphs presented in following figure panels: 3B, 6B, 7A, 7C, S2A, S2B, S9, and S10B.

5) The image of the patient 1-derived fibroblast stained with mAb414 in Figure 4 suggests that the distribution of NPCs differs between this cell and the control 1 fibroblast. However, the images of mAb414-stained NPCs and their corresponding quantifications shown in Figure S6A do not suggest that patient-derived fibroblasts have a defect in NPC distribution. Thus, the authors should select a more representative image for Patient 1 in Figure 4.

6) Given the large difference in LAP1B and LAP1C expression levels obtained following lentiviral transduction of the patient-derived fibroblasts shown in Figure S10 and the minimal rescue of the cellular phenotypes observed in the LAP1C expressing patient-derived fibroblasts reported in Figure 7, it seems strange to me that the authors chose to include the individual rescue experiments in the main figures of the manuscript, while the more impressive rescue experiments where both LAP1 isoforms were expressed in the patient-derived fibroblasts are relegated to the Supplemental material. I would strongly suggest that the authors combine the dual LAP1 isoform rescue experiments with the single isoform rescue experiments in Figure 7. The Western blot can remain in Figure S10.

Reviewer #4 (Remarks to the Author):

The authors have constructively addressed our concerns and the new data added in revision are in

excellent agreement with their original data and interpretations. Moreover, all of the stylistic/citation changes/additions suggested by us and the other reviewers were implemented in the revised version. This significant body of work is well executed, novel, and has major implications for our understanding of human disease in the context of the nuclear envelope. Consequently, we recommend timely publication of the manuscript in its present form.

Point-by-Point Response

General summary of changes in the figures

Panel c in Figure 1 has been replaced with a different image, following further discussion with the patient families, in accordance with the journal's guidelines.

Two panels in Figure 4 have been replaced with different images - see below.

Supplementary Figure 4 has been **cancelled** and all the data (EM images of control & patient fibroblasts) has been merged into **Figure 5**, making it larger.

Panel d has been added to **Figure 7**, containing data (combined rescue with LAP1B & LAP1C) previously shown in Supplementary Figure 10b.

The number of **Supplementary Figures** has been reduced from 10 to 9.

Supplementary Figure 9 now contains only the supporting Western blot information for Figure 7, that appeared in previous Supplementary Figure 10a.

* **Please note** that some tracked changes in the manuscript text and figure legends and in the figures (e.g., change in title, shortened subheadings, removed scale bar labels, dot plots overlaid on bar charts) have been made in response to editorial requests.

Response to Reviewer 2

Reviewer #2 (Remarks to the Author):

Overall, I feel that the authors have successfully addressed my previously raised concerns. That being said, I do have a few remaining suggestions that I would like to see addressed before this manuscript can be accepted for publication.

1) In response to the authors' response to my 5th Major Concern, they state "adding EM images of control nuclei to Figure 5 itself would distract from the main issue". I strongly disagree with this statement, as including the control images next to the patient fibroblasts is critical for comparing the nuclear envelope defects observed in the absence of LAP1B and LAP1C.

All the EM images formerly included in **Supplementary Figure 4** have been merged into **Figure 5**, making it larger and now containing panels a-c. Accompanying (tracked) changes in the Results section appear on pages 11-12.

2) The authors should change the title "Transfection of LAP1-coding constructs into patient-derived fibroblasts rescues multiple cellular phenotypes" to "Expression of LAP1-encoding constructs into patient-derived fibroblasts rescues multiple cellular phenotypes". They are using lentiviral transduction, not transfection, to express the LAP1-encoding constructs in the patient-derived fibroblasts.

The title of this subheading has been shortened, to <60 characters with spaces, according to the editorial requests and could not include this information. However, we did change the title of **Figure 7** accordingly and transduction is mentioned in the text and in the Methods subheading: "Lentiviral transduction" (page 28).

3) In the Discussion, the authors state that torsinA is "another direct interaction of the LINC complex"; however, this statement is not correct. While torsinA was shown to have affinity for the KASH domains of nesprins 1-3 by Nery et al. (2008) J Cell Sci, this does not mean that torsinA can interact with an assembled LINC complex. To my knowledge, no one has ever shown that torsinA can directly interact with a LINC complex.

The text on page 18 has been changed to: "LAP1 binds and activates the ATPase activity of torsinA, while torsinA has been shown to bind the KASH domains of LINC complex components^{21,22}." However, we think that the statement that follows this sentence, regarding a potential NE toolbox protein, still holds true.

4) The authors should indicate the statistical significance, or lack thereof, in the graphs presented in following figure panels: 3B, 6B, 7A, 7C, S2A, S2B, S9, and S10B.

The statistical information has been added to all the relevant figures. Note that former Supplementary Figure 10b is now included in the main figures as **Figure 7d**.

5) The image of the patient 1-derived fibroblast stained with mAb414 in Figure 4 suggests that the distribution of NPCs differs between this cell and the control 1 fibroblast. However, the images of mAb414-stained NPCs and their corresponding quantifications shown in Figure S6A do not suggest that patient-derived fibroblasts have a defect in NPC distribution. Thus, the authors should select a more representative image for Patient 1 in Figure 4.

Indeed, (new) **Supplementary Figure 5a,b** clearly shows there is no difference in NPC distribution. The images in **Figure 4** should not be used to judge NPC distribution and we have now **added a sentence** to the end of its legend, referring the reader to

Supplementary Figure 5 for this purpose. The NPCs are clearly not in focus in the single optical sections shown in Figure 4, as opposed to the upper tangential sections shown in Supplementary Figure 5a. We cannot replace the **Patient 1** mid-section mAb414 panel used for the z-stack in Figure 4 with a tangential section. That would defeat the whole purpose in this figure, which is showing that the cytoplasmic channel can be followed through the nucleus from top-to-bottom (the single section should represent the stack, not its end). Instead, we replaced the single section panels for **Control 1** with a mid-section, so that the overall mAb414 staining pattern is similar in the patient and control images.

6) Given the large difference in LAP1B and LAP1C expression levels obtained following lentiviral transduction of the patient-derived fibroblasts shown in Figure S10 and the minimal rescue of the cellular phenotypes observed in the LAP1C expressing patient-derived fibroblasts reported in Figure 7, it seems strange to me that the authors chose to include the individual rescue experiments in the main figures of the manuscript, while the more impressive rescue experiments where both LAP1 isoforms were expressed in the patient-derived fibroblasts are relegated to the Supplemental material. I would strongly suggest that the authors combine the dual LAP1 isoform rescue experiments with the single isoform rescue experiments in Figure 7. The Western blot can remain in Figure S10.

The figures have now been changed according to this suggestion: former Supplementary Figure 10b is the new **Panel d** in **Figure 7**, containing the combined rescue with both isoforms. The new **Supplementary Figure 9** now contains only the supporting Western blot information. Accompanying (tracked) changes in the Results section appear on pages 16-17.

We thank all of the reviewers for their comments and hope that the manuscript will now be found acceptable for publication in *Nature Communications*.